# Simultaneous impairment of neuronal and metabolic function of mutated gephyrin in a patient with epileptic encephalopathy

Borislav Dejanovic[1,*,‡,§], Tania Djémié[2,3,‡], Nora Grünewald[1], Arvid Suls[2,3,4], Vanessa Kress[1], Florian Hetsch[5], Dana Craiu[6,7], Matthew Zemel[8], Padhraig Gormley[9,10,11,12], Dennis Lal[13,14], EuroEPINOMICS Dravet working group[†], Candace T Myers[8], Heather C Mefford[8], Aarno Palotie[9,12,15,16,17,18,19], Ingo Helbig[20,21], Jochen C Meier[5], Peter De Jonghe[2,3,22], Sarah Weckhuysen[2,3,23,24,**,§] & Guenter Schwarz[1,10,11,***,§]

## Abstract

Synaptic inhibition is essential for shaping the dynamics of neuronal networks, and aberrant inhibition plays an important role in neurological disorders. Gephyrin is a central player at inhibitory postsynapses, directly binds and organizes GABA$_A$ and glycine receptors (GABA$_A$Rs and GlyRs), and is thereby indispensable for normal inhibitory neurotransmission. Additionally, gephyrin catalyzes the synthesis of the molybdenum cofactor (MoCo) in peripheral tissue. We identified a *de novo* missense mutation (G375D) in the gephyrin gene (*GPHN*) in a patient with epileptic encephalopathy resembling Dravet syndrome. Although stably expressed and correctly folded, gephyrin-G375D was non-synaptically localized in neurons and acted dominant-negatively on the clustering of wild-type gephyrin leading to a marked decrease in GABA$_A$R surface expression and GABAergic signaling. We identified a decreased binding affinity between gephyrin-G375D and the receptors, suggesting that Gly375 is essential for gephyrin–receptor complex formation. Surprisingly, gephyrin-G375D was also unable to synthesize MoCo and activate MoCo-dependent enzymes. Thus, we describe a missense mutation that affects both functions of gephyrin and suggest that the identified defect at GABAergic synapses is the mechanism underlying the patient's severe phenotype.

1  Department of Chemistry, Institute of Biochemistry, University of Cologne, Cologne, Germany
2  Neurogenetics Group, Department of Molecular Genetics, VIB, Antwerp, Belgium
3  Laboratory of Neurogenetics, Institute Born-Bunge, University of Antwerp, Antwerp, Belgium
4  GENOMED, Center for Medical Genetics, University of Antwerp, Antwerp, Belgium
5  Division Cell Physiology, Zoological Institute, Technische Universität Braunschweig, Braunschweig, Germany
6  Pediatric Neurology Clinic, Al Obregia Hospital, Bucharest, Romania
7  Department of Neurology, Pediatric Neurology, Psychiatry, Child and Adolescent Psychiatry, and Neurosurgery, Carol Davila University of Medicine and Pharmacy, Bucharest, Romania
8  Division of Genetic Medicine, Department of Pediatrics, University of Washington, Seattle, WA, USA
9  Wellcome Trust Sanger Institute, Wellcome Trust Genome Campus, Hinxton, UK
10 Center for Molecular Medicine Cologne (CMMC), University of Cologne, Cologne, Germany
11 Cologne Excellence Cluster on Cellular Stress Responses in Aging-Associated Diseases (CECAD), University of Cologne, Cologne, Germany
12 Psychiatric & Neurodevelopmental Genetics Unit, Department of Psychiatry, Massachusetts General Hospital and Harvard Medical School, Boston, MA, USA
13 Cologne Center for Genomics, Cologne Excellence Cluster on Cellular Stress Responses in Aging-Associated Diseases (CECAD), University of Cologne, Cologne, Germany
14 Department of Neuropediatrics, University Medical Faculty Giessen and Marburg, Giessen, Germany
15 Institute for Molecular Medicine Finland (FIMM), University of Helsinki, Helsinki, Finland
16 Program in Medical and Population Genetics, The Broad Institute of MIT and Harvard, Cambridge, MA, USA
17 The Stanley Center for Psychiatric Research, The Broad Institute of MIT and Harvard, Cambridge, MA, USA
18 Analytic and Translational Genetics Unit, Department of Medicine, Massachusetts General Hospital and Harvard Medical School, Boston, MA, USA
19 Department of Neurology, Massachusetts General Hospital, Boston, MA, USA
20 Department of Neuropediatrics, University Medical Center Schleswig-Holstein, Christian Albrechts University, Kiel, Germany
21 Division of Neurology, The Children's Hospital of Philadelphia, Philadelphia, PA, USA
22 Division of Neurology, Antwerp University Hospital, Antwerp, Belgium
23 Inserm U 1127, CNRS UMR 7225, Sorbonne Universités, UPMC Univ Paris 06 UMR S 1127, Institut du Cerveau et de la Moelle épinière, ICM, Paris, France
24 Centre de reference épilepsies rares, Epilepsy unit, AP-HP Groupe hospitalier Pitié-Salpêtrière, F-75013, Paris, France
   *Corresponding author. Tel: +1 650 333 4376; E-mail: b.dejanovic@uni-koeln.de
   **Corresponding author. Tel: +32 32 65 1022; Fax: +32 32 65 1112; E-mail: sarahweck@hotmail.com
   ***Corresponding author. Tel: +49 221 470 6440; Fax: +49 221 470 5092; E-mail: gschwarz@uni-koeln.de
   †EuroEPINOMICS Dravet working group: B.P.C. Koeleman; E.H. Brilstra; S. Sisodiya; S. Baulac; C. Depienne; J. Serratosa; P. Striano; C. Marini; R. Guerrini; H. Caglayan; T. Talvik; D. Hoffman; S. von Spiczak; J. Jähn
   ‡These authors contributed equally to this work
   §Joint senior authors

  

**Keywords** Dravet syndrome; epileptic encephalopathy; GABA$_A$ receptors; gephyrin; molybdenum cofactor

**Subject Categories** Genetics, Gene Therapy & Genetic Disease; Neuroscience

## Introduction

With a burden of 68 million people affected worldwide, epilepsy is one of the most common neurological disorders (Ngugi et al, 2010). The disease is characterized by recurrent spontaneous seizures which are provoked by a disturbed balance between excitatory and inhibitory cerebral activity leading to hyperexcitability (Schwartzkroin, 2012). Epileptic encephalopathies (EEs) are very severe forms of epilepsy, which are associated with cognitive impairment and usually have an early onset (Cross & Guerrini, 2013). One of the best-defined phenotypes within the EEs is Dravet syndrome. Patients present with febrile seizures in the first year of life, and as the disease progresses, other seizure types such as myoclonic and tonic–clonic seizures become more prominent. In most patients, the epilepsy is refractory and developmental delay occurs soon after seizure onset (Dravet, 2011). Besides structural and metabolic defects, EEs can be caused by genetic alterations. Most of these EEs present as monogenic disorders due to de novo mutations (EuroEPINOMICS-RES Consortium et al, 2014; Allen et al, 2013; Veeramah et al, 2013). Patients with EEs are severely disabled and therefore mostly present as sporadic cases, not transmitting mutations to next generations. About 80% of patients with Dravet syndrome, for example, carry a de novo mutation in the gene SCN1A (Parihar & Ganesh, 2013). In this study, we performed whole exome sequencing (WES) on a patient with a Dravet-like syndrome (not carrying an SCN1A mutation) and his parents and identified a heterozygous de novo missense mutation in the gephyrin gene (GPHN).

Gephyrin is the major postsynaptic scaffolding protein at inhibitory synapses (Fritschy et al, 2008). It directly interacts with subunits of glycine and GABA$_A$ receptors (GlyRs and GABA$_A$Rs), major components of fast inhibitory transmission, and regulates clustering and diffusion of these receptors (Luscher et al, 2011; Choquet & Triller, 2013). Gephyrin forms multimeric complexes at postsynaptic membranes (Dejanovic et al, 2014b; Tyagarajan & Fritschy, 2014), which is essential for normal inhibitory signaling (Calamai et al, 2009). The protein is composed of three functional domains (Fig 1A): (i) a N-terminal G-domain that facilitates gephyrin trimerization; (ii) a central (C-)domain that is highly modified by posttranslational regulation and interacts with a number of neuronal proteins; and (iii) a C-terminal E-domain that binds to GlyRs and GABA$_A$Rs using a common binding site (Tyagarajan & Fritschy, 2014). Besides its synaptic function, gephyrin has a second, metabolic function: it catalyzes the last two steps in the biosynthesis of the highly conserved molybdenum cofactor (MoCo), a reaction that requires the G- and E-domain of gephyrin (Belaidi & Schwarz, 2013). MoCo in humans is required for the activity of four molybdo enzymes that catalyze redox reactions (Schwarz et al, 2009). MoCo deficiency in humans is a severe autosomal recessive metabolic disorder characterized by untreatable neonatal seizures starting at birth, neuronal loss and ultimate death in the first years of life, mainly caused through accumulation of toxic sulfite due to the loss of activity of sulfite oxidase (Schwarz et al, 2009).

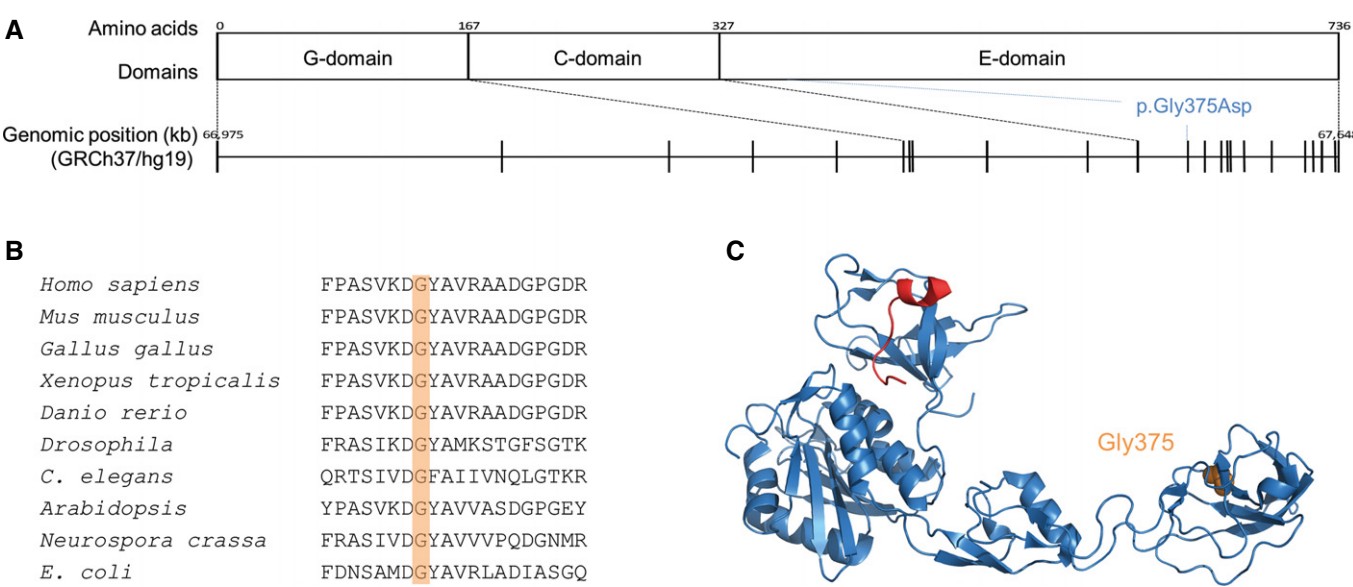

**Figure 1. Localization of the identified GPHN mutation (NM_001024218; NP_001019389).**

A   Protein and genetic structure including the different gephyrin domains and genomic location of the gene with an indication of the different exons. The position of the mutation identified in this study is indicated on the protein level, affecting the E-domain and on the genomic level, affecting exon 12.

B   Multiple alignment of gephyrin and homologous proteins from various species from different kingdoms of life (aligned using Clustal Omega).

C   Co-crystal structure of gephyrin E-domain (blue) and bound GlyR β-loop (red, PDB code 2FTS (Kim et al, 2006)). The mutated gephyrin Gly375 residue is shown in orange.

Given the importance of gephyrin in the organization of inhibitory synapses, it is not surprising that gephyrin dysfunction can lead to epilepsy. Previously, we have identified stress-induced irregular splicing of *GPHN* resulting in the expression of truncated gephyrin variants in patients with temporal lobe epilepsy (Forstera *et al*, 2010). More recently, we and others have identified large deletions in *GPHN* in patients with different neurological disorders such as idiopathic generalized epilepsy (IGE), autism, schizophrenia and seizures (Lionel *et al*, 2013; Dejanovic *et al*, 2014a). In the present study, we identified a *de novo* heterozygous missense mutation in *GPHN* in a patient with Dravet-like syndrome. Our subsequent functional characterization of this novel gephyrin mutation identified the first alteration in gephyrin that disrupts two of its primary functions in a residue-specific manner. In addition, our study provides novel insights into the mechanism of gephyrin-mediated clustering of inhibitory neuroreceptors.

## Results

### Whole exome sequencing and follow-up screenings

Whole exome sequencing of a patient with Dravet-like syndrome and his unaffected parents and downstream analysis only revealed a single *de novo* variant: a heterozygous missense mutation, c.1,124G>A, p.Gly375Asp in *GPHN* (NM_001024218, Fig 1A). We validated the mutation and confirmed the *de novo* status using classical Sanger sequencing. The nucleotide and the corresponding amino acid are both highly conserved (GERP: 5.77; Fig 1B). The mutation is not present in control databases (1000 Genomes Project, Exome Variant Server, dbSNP, in-house data), and only two other missense variants in the same exon have been detected in > 33,600 European individuals (ExAc Browser). The p.Gly375Asp mutation is predicted to be damaging by prediction programs (Polyphen-2: probably damaging [1], SIFT: damaging [0], MutationTaster: disease causing [1]). On a structural level, the mutation is located in the E-domain (Fig 1C). Recessive analyses of the data did not lead to the identification of any recessive variants in the patient. The screening of follow-up cohorts using the Multiplex Amplification of Specific Targets for Resequencing (MASTR) and the molecular inversion probes (MIP) technology did not lead to the identification of additional cases with mutations in *GPHN*.

### Clinical description of the patient with a *de novo* gephyrin-G375D mutation

The patient is a 21-year-old man, born from non-consanguineous parents from Caucasian origin. He was born at 39 weeks after a spontaneous delivery following an imminent abortion at 38 weeks. Birth parameters were normal as was development during the first year of life. Head control was present at 4 months, he sat independently at 8 months, and he walked at 1 year and 3 months. At the age of 11 months, he had his first simple febrile seizure. Since then, he showed generalized tonic–clonic seizures, myoclonic seizures, right or left hemiclonic seizures, atypical absences, and focal dyscognitive seizures, all with or without fever. Phenobarbital was ineffective, but seizures were eventually controlled by carbamazepine. Every attempt to stop or reduce the medication led to

seizure recurrence. EEG during the disease course showed focal sharp slow waves right or left and multifocal discharges. Soon after epilepsy onset, developmental delay was noted. At the age of 18 years, he had severe intellectual disability (ID) with an IQ of 33. MRI of the brain and neurological examination were normal. This patient was diagnosed with Dravet-like syndrome.

### Gephyrin-G375D mutation alters gephyrin distribution and clustering in neurons

Malfunction of either of the two known functions of gephyrin could contribute to seizures and epileptogenesis (Forstera *et al*, 2010; Reiss *et al*, 2001; Dejanovic *et al*, 2014a). To evaluate whether the mutation influences the postsynaptic clustering of gephyrin, we expressed GFP-tagged wild-type (WT) gephyrin or gephyrin carrying the p.Gly375Asp mutation (termed GFP-G375D) in primary hippocampal neurons and evaluated its cellular distribution after four days of expression (Fig 2A). GFP-gephyrin formed postsynaptic clusters along the dendrites and cell body. Strikingly, GFP-G375D was diffusively distributed in neurons and filled all cellular compartments including spine heads, the morphological structures of excitatory synapses (Fig 2A). Only a minor fraction of GFP-G375D formed clusters (Fig 2B), which, however, were mainly non-synaptically localized and did not co-localize with the presynaptic marker protein VGAT (Fig 2C and D). Together, these results suggest that the mutation leads to loss-of-clustering function of the mutant gephyrin.

Given that gephyrin oligomerizes at postsynaptic sites, we wondered whether gephyrin-G375D influences the clustering of WT-gephyrin. We used a well-established monoclonal antibody against gephyrin (3B11) (Smolinsky *et al*, 2008), which did not recognize GFP-G375D in immunocytochemistry and Western blot assays (Fig EV1A and B), and was therefore used to selectively detect endogenous gephyrin in hippocampal neurons (Fig 3A). Compared to non-transfected neighboring neurons or GFP-expressing control neurons, the number of endogenous gephyrin clusters in GFP-G375D-expressing neurons was significantly decreased (Fig 3B). Additionally, remaining endogenous gephyrin clusters were significantly decreased in size (Fig 3C), suggesting that GFP-G375D acts dominant-negatively on the WT protein in the patient, thereby probably impairing scaffold formation at inhibitory synapses.

### Gephyrin-G375D mutation impairs GABA$_A$R clustering and signaling

Postsynaptic gephyrin clustering is essential for normal anchoring and dynamics of GABA$_A$Rs. We used cell-surface immunostaining of living hippocampal neurons co-expressing myc-tagged WT-gephyrin or G375D and pHluorin-tagged GABA$_A$R $\gamma$2 subunit (Reinthaler *et al*, 2015) (Fig 4A). The size and fluorescence intensity of cell-surface-expressed $\gamma$2-containing GABA$_A$R clusters were significantly decreased in myc-G375D-expressing neurons compared to neurons expressing myc-WT-gephyrin (Fig 4A–C). The fluorescence of total $\gamma$2 subunits (as measured by pHluorin fluorescence) remained unchanged (Fig 4A and D). The presence of myc-G375D, however, did not influence the density of surface $\gamma$2 clusters (Fig 4E).

To assess the functional consequence of the G375D substitution on GABAergic synapses, we recorded miniature inhibitory

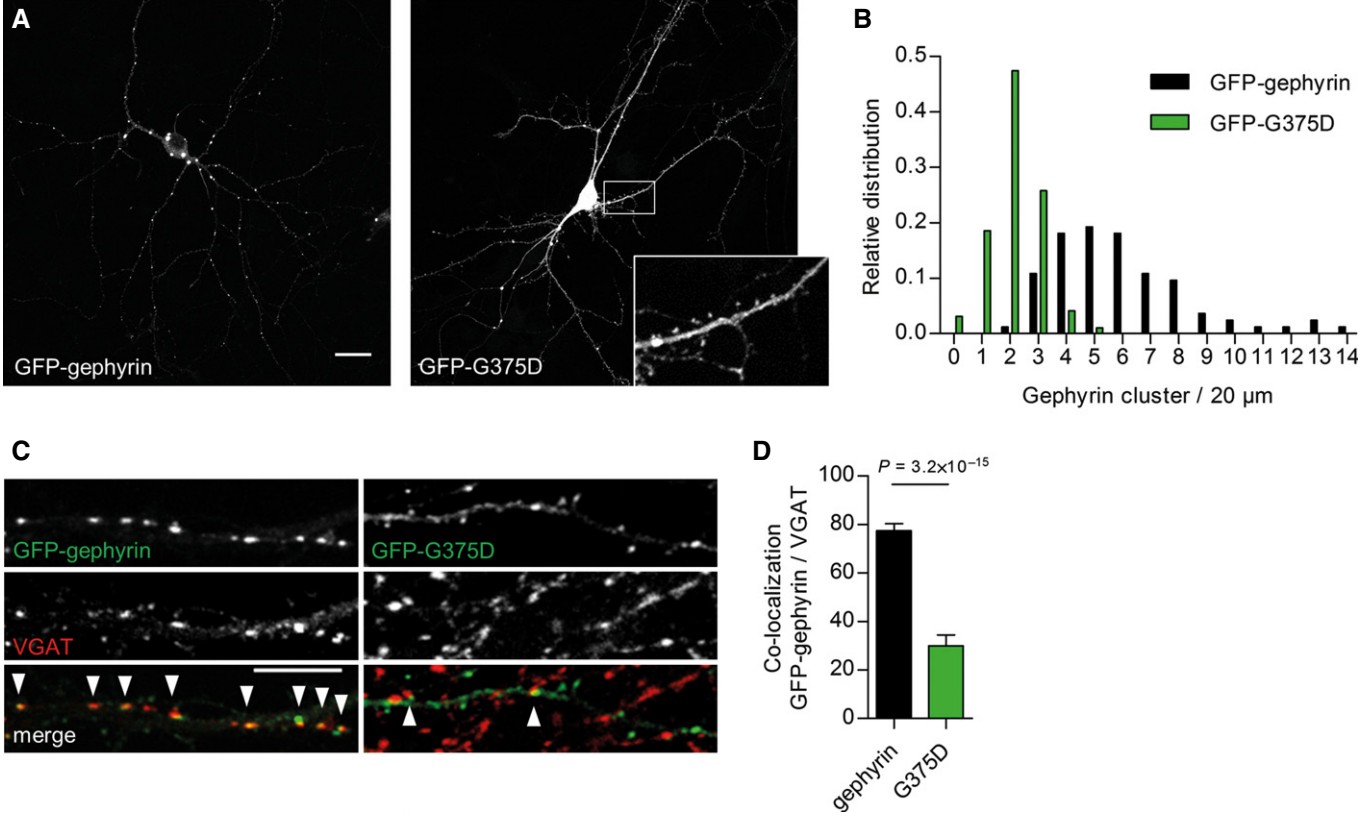

**Figure 2. Expression of WT-gephyrin and gephyrin-G375D in primary neurons.**

A   Representative images of GFP-gephyrin and GFP-G375D, expressed in hippocampal neurons at 9 + 4 days *in vitro*. Inset shows higher magnification. Scale bar, 20 μm.

B   Relative distribution of cluster density of gephyrin variants on dendrites from transfected neurons. Number of gephyrin clusters in each dendrite is normalized to 20 μm segments.

C   Representative segments of dendrites expressing GFP-gephyrin or GFP-G375D and immunostained with the presynaptic marker VGAT. Arrowheads show co-localization of gephyrin and VGAT clusters. Scale bar, 10 μm.

D   Quantification of co-localization between gephyrin and VGAT clusters (gephyrin-VGAT 77.4 ± 2.9%, G375D-VGAT 30 ± 4.4%). 31 GFP-gephyrin and 30 GFP-G375D neurons from three independent cultures were used for quantifications. Results are expressed as mean ± SEM and data were analyzed by *t*-test.

postsynaptic currents (mIPSCs) in hippocampal neurons expressing GFP-gephyrin or GFP-G375D (Fig 4F). Mean mIPSC amplitudes as well as the median frequency were significantly reduced in neurons expressing GFP-G375D as compared to neurons expressing GFP-gephyrin (Fig 4G and H). In line with the neuronal cell-surface staining, decreased amplitude in gephyrin-G375D-expressing neurons can be attributed to a decreased number of postsynaptic GABA$_A$Rs. The reduction in frequency may reflect a loss of GABA$_A$Rs at a subgroup of inhibitory postsynapses.

Taken together, our cellular and electrophysiological data suggest that the G375D substitution abolishes postsynaptic clustering of gephyrin, which exerts dominant-negative effects on GABA-ergic postsynaptic protein scaffolds and GABA$_A$R signaling.

### Gephyrin-G375D is properly folded and stably expressed in eukaryotic cells

We further wanted to understand the underlying mechanism of impaired gephyrin-G375D postsynaptic clustering. We analyzed whether gephyrin-G375D was able to interact with collybistin, a

RhoGEF protein essential for membrane trafficking and postsynaptic clustering of gephyrin (Papadopoulos *et al*, 2007). Upon co-expression of GFP-gephyrin or GFP-G375D with mCherry-tagged collybistin II (CBII) in HEK293 cells, similar amounts of CBII were co-immunoprecipitated using GFP antibodies (Fig EV2A). Furthermore, CBII co-localized with GFP-gephyrin and GFP-G375D in COS7 cells and induced the formation of submembranous gephyrin micro-clusters (Fig EV2B and C). Although we noted that compared to GFP-gephyrin, less GFP-G375D-expressing cells formed CBII-induced submembranous microclusters (Fig EV2C), we conclude that this effect is not responsible for the severe cluster deficiency observed in neurons and gephyrin–CBII interaction seemed to be intact in the mutant.

We next expressed GFP-G375D in HEK293 cells where expression levels were comparable to GFP-gephyrin and no degradation was detected, suggesting that gephyrin-G375D is expressed as a stable and properly folded protein in eukaryotic cells (Fig 5A). In line with these results, the folding and secondary structure of 6His-tagged gephyrin-G375D, expressed and purified from *E. coli* cells, was very similar to that of 6His-tagged gephyrin as measured by circular

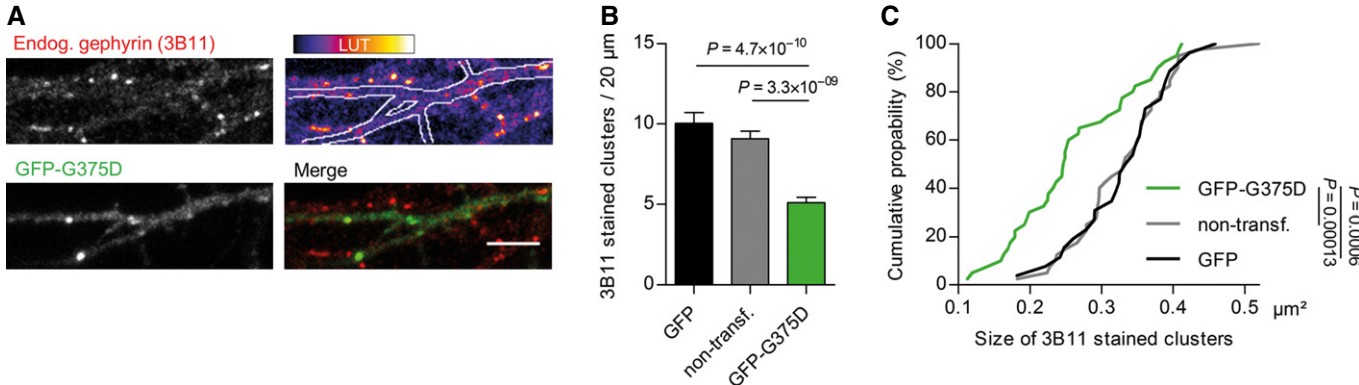

**Figure 3.  Gephyrin-G375D acts dominant-negatively on postsynaptic clustering of WT gephyrin in neurons.**

A    Representative dendritic segment of a neuron expressing GFP-G375D and 3B11-immunostained endogenous gephyrin (red). Scale bar, 5 μm.

B, C    Quantification of (B) density and (C) size of endogenous gephyrin clusters in neurons expressing GFP, GFP-G375D, or non-transfected neighboring neurons. 20 GFP-G375 and non-transfected neurons and 15 GFP-expressing neurons from three independent cultures were used for quantifications. Results are expressed as mean ± SEM, and data were analyzed by *t*-test.

dichroism (CD) spectroscopy (Fig 5B). We next wondered whether oligomerization of gephyrin is affected by the mutation and performed size-exclusion chromatography with purified full-length gephyrin and isolated gephyrin E-domain. Both 6His-gephyrin and 6His-G375D eluted at the size corresponding to a trimer (elution approximated 250 kDa, monomer is 83 kDa) and the E-domain as dimers (elution approximated 105 kDa, monomer is 48 kDa, Fig 5C), which is in agreement with the literature (Saiyed *et al*, 2007). Also, when we co-expressed myc-tagged WT-gephyrin with GFP-gephyrin or GFP-G375D in HEK293 cells, similar amounts of GFP-tagged gephyrins were co-immunoprecipitated with myc-specific antibodies (Fig 5D), suggesting that self-oligomerization of gephyrin was intact (Fig 5D).

**Gephyrin-G375D binds GABA$_A$R and GlyR with reduced affinity**

As the biophysical and biochemical properties of gephyrin-G375D were not notably impaired by the mutation, we speculated that the mutation might influence the interaction between gephyrin and the inhibitory neuroreceptors. Although Gly375 is not localized in the known binding site for GlyRs and GABA$_A$Rs (Fig 1C), gephyrin–receptor interaction might be more complex than initially anticipated (Maric *et al*, 2014b). We therefore wondered whether the binding toward GABA$_A$Rs is decreased. We synthesized a biotinylated peptide containing residues 373–424 of the cytoplasmic loop of the GABA$_A$R α3-subunit (Fig 6A), which has been shown to have the highest binding affinity among the gephyrin-interacting GABA$_A$R subunits (Maric *et al*, 2011). The peptide was immobilized to NeutrAvidin beads and used to pull-down purified 6His-tagged gephyrin or 6His-tagged G375D. Compared to 6His-tagged gephyrin, a significantly smaller fraction of 6His-tagged G375D was co-sedimented by the α3-loop (Fig 6B). To confirm these qualitative results, we performed surface plasmon resonance (SPR)-based interaction studies (Kowalczyk *et al*, 2013) and coupled the biotinylated α3-loop to a streptavidin SPR sensor chip. Purified 6His-tagged gephyrin and 6His-tagged G375D were passed over the chip at increasing concentrations, and response units derived from interacting proteins were determined (Fig 6C). The dissociation constants

($K_D$) obtained from three independent experiments for WT-gephyrin and gephyrin-G375D were 3.0 ± 0.9 and 7.8 ± 1.4 μM, respectively, demonstrating a significantly reduced affinity for the gephyrin-G375D variant ($n = 3$, $P = 0.032$, *t*-test, Fig 6D). The difference in the maximal response units (Fig 6D; WT-gephyrin 3,426 ± 556, gephyrin-G375D 965 ± 403, $n = 3$, $P = 0.021$, *t*-test) further showed that the amount of bound 6His-tagged G375D was less than 30% of that of 6His-gephyrin, which mirrors the results from our pull-down experiments.

Knowing that all GABA$_A$R and GlyR subunits share a common binding site on gephyrin (Maric *et al*, 2011; Kowalczyk *et al*, 2013; Fig 1C), we next asked whether the mutation also impairs the interaction with GlyRs. We used isothermal titration calorimetry (ITC) to measure the binding affinity between gephyrin and the cytoplasmic loop of the GlyR β-subunit (GlyR-βL), which binds gephyrin with high affinity (Schrader *et al*, 2004; Maric *et al*, 2011). As reported previously, we identified two binding sites between GlyR-βL and 6His-gephyrin: one high-affinity binding site with a $K_D$ of 0.02 ± 0.004 μM and one low-affinity binding site with a $K_D$ of 2.93 ± 0.49 μM (Fig 6E) (Specht *et al*, 2011; Herweg & Schwarz, 2012). Interestingly, for 6His-tagged G375D, we only detected one binding event with a $K_D$ of 0.39 ± 0.10 μM, suggesting that the mutation leads to the loss of the low-affinity binding site and affects the binding affinity of the high-affinity binding site. Moreover, the decreased stoichiometry of 6His-G375D/GlyR-βL interaction suggests a much lower occupancy for gephyrin-G375D compared to WT-gephyrin (Fig 6E). Together, these results suggest that Gly375 is essential for gephyrin–receptor interaction and the replacement of Gly375 to aspartate severely impairs the interaction between gephyrin and GABA$_A$Rs/GlyRs.

**Gephyrin-G375D lacks MoCo synthesis activity**

As gephyrin exhibits a second, equally crucial, metabolic function catalyzing the last two steps of MoCo biosynthesis (Fig 7A), we wondered if this catalytic function is also influenced by the G375D mutation. In the absence of any biological samples from the patient, we used an *in vitro* assay that we recently developed to measure the

    

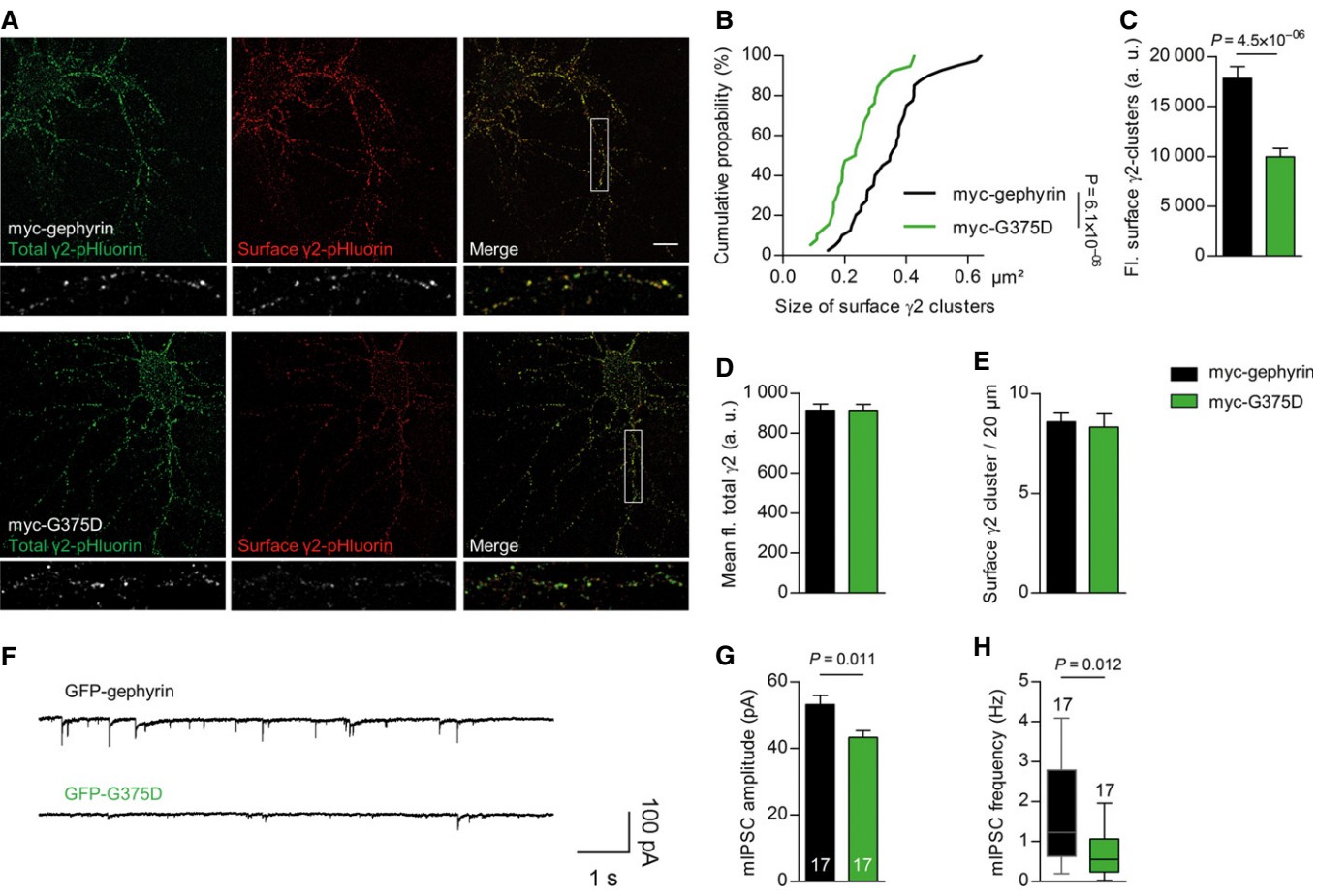

**Figure 4. Gephyrin-G375D decreases cell-surface expression of GABA$_A$Rs and impairs postsynaptic currents.**

A    Representative neurons expressing myc-tagged gephyrin or G375D and pHluorin-tagged GABA$_A$R γ2. Surface-expressed γ2 subunits were immunolabeled with GFP antibodies in red, and the pHluorin fluorescence (total γ2 levels) is shown in green.

B, C    Size (B) and fluorescence (fl.) intensity (C) of surface-expressed γ2 clusters in myc-tagged gephyrin and myc-G375D-expressing dendrites (*n* = 20 myc-gephyrin and 21 myc-G375D neurons in B and 18 myc-gephyrin- and myc-G375D neurons in C).

D    Mean γ2-pHluorin fluorescence in myc-gephyrin- and myc-G375D-expressing dendrites (*n* = 20 myc-tagged gephyrin- and 18 myc-G375D-expressing neurons).

E    Number of surface γ2 clusters on dendrites normalized to 20 μm segments (*n* = 18 myc-tagged gephyrin- and myc-G375D-expressing neurons).

F    Representative traces of miniature recordings on cultured hippocampal neurons transfected with either GFP-gephyrin or GFP-G375D.

G    mIPSC amplitudes from neurons expressing GFP-gephyrin or GFP-G375D (GFP-gephyrin 53.11 ± 2.8; GFP-G375D 43.25 ± 2.0 pA). Numbers show analyzed cell numbers. Data (mean +/− SD) were analyzed by one-way ANOVA and *post hoc* Tukey test.

H    Frequency of the mIPSCs from neurons expressing GFP-gephyrin or GFP-G375D. The horizontal line within the boxes indicates the median frequency for the represented condition (bottom/top boundary indicates the 25th/75th percentile). Top and bottom error bars indicate the 90th and 10th percentiles, respectively. Analysis by Mann–Whitney test.

Data information: In (B–E), at least three independent cultures were used for each experiment and results are expressed as mean ± SEM; data were analyzed by *t*-test.

amount of MoCo produced by purified gephyrins (Belaidi & Schwarz, 2013). While 6His-tagged gephyrin was able to synthesize MoCo, 6His-G375D was enzymatically inactive and unable to catalyze MoCo biosynthesis (Fig 7B). The quantity of MoCo produced by 6His-G375D was similar to that of 6His-tagged gephyrin-D580A, a gephyrin mutation which was identified homozygously in a patient with MoCo deficiency (Reiss *et al*, 2011). Interestingly, GFP-tagged gephyrin-D580A expressed in hippocampal neurons formed post-synaptic clusters and was indistinguishable from WT-gephyrin, suggesting that both gephyrin functions act mutually independent (Fig EV3). When 6His-gephyrin and 6His-G375D were pre-incubated in a 1:1 ratio, 53.4 ± 4.95% of gephyrin-catalyzed MoCo was synthesized (Fig 7C), suggesting that, at least *in vitro*, gephyrin-G375D does

not act dominant-negatively on gephyrin's MoCo function. Taken together, the *GPHN* mutation G375D identified in a patient with Dravet-like syndrome abolished the synaptic as well as the metabolic function of gephyrin.

# Discussion

We present the first patient with an infantile onset EE and a heterozy-gous *de novo* mutation in the *GPHN* gene. In this unique case, the missense mutation c.1,124G>A, p.G375D (NM_001024218), abol-ishes both functions of gephyrin without affecting the structure and folding of the protein: (i) gephyrin-G375D postsynaptic clustering

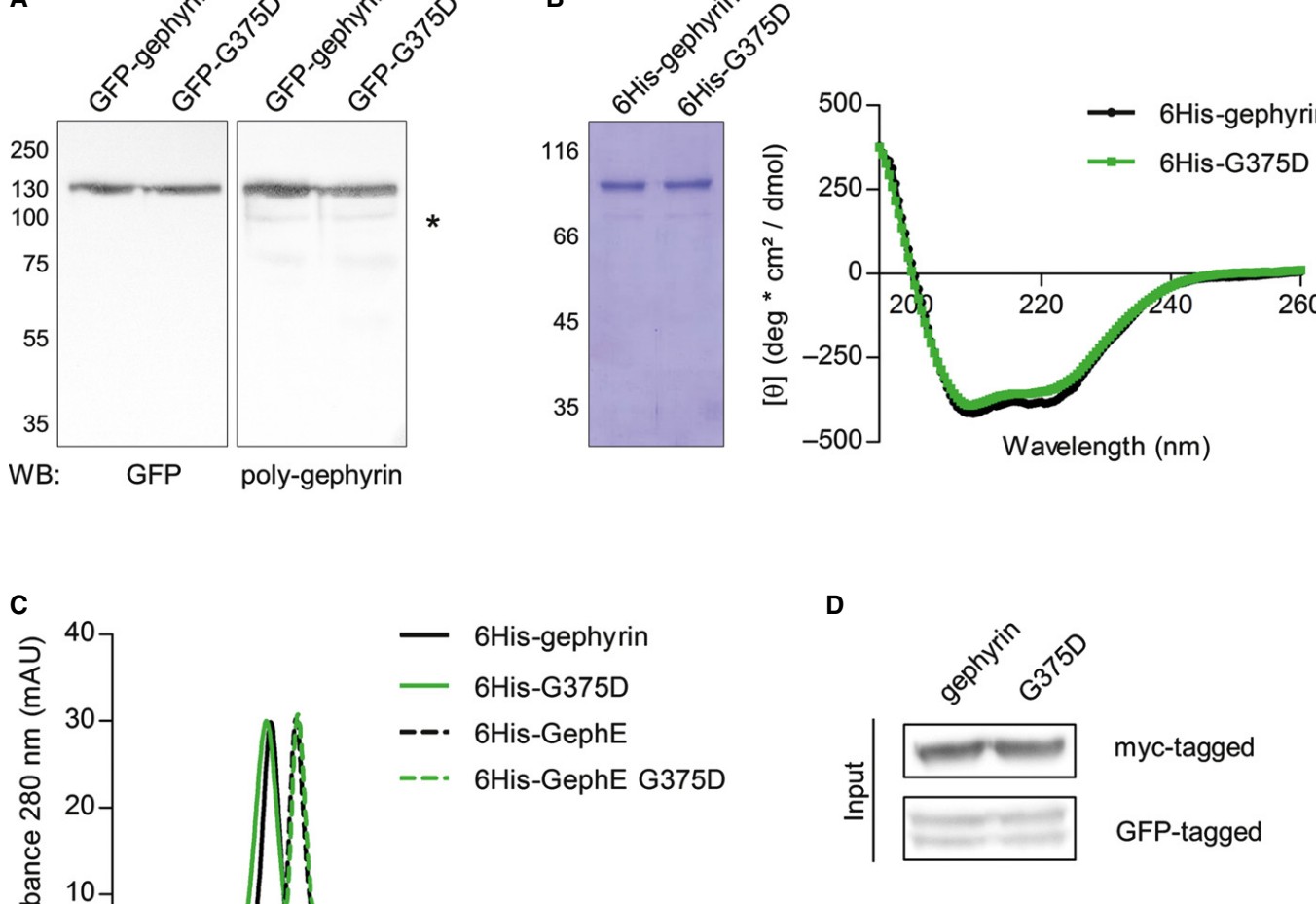

**Figure 5. Gephyrin-G375D is stably expressed and oligomerizes with WT-gephyrin.**

A  Western blot of HEK293 cells expressing GFP-tagged gephyrin or G375D using GFP-specific and polyclonal gephyrin antibodies. Note that both gephyrin variants are stably expressed and show no degradation in the cells. Asterisk in the poly-gephyrin blot shows endogenous gephyrin.

B  SDS–PAGE of *E. coli*-expressed and purified 6His-tagged gephyrins and CD spectra. Mean residue ellipticity of gephyrin and G375D shows profiles with minima at 208 nm and 222 nm.

C  Size-exclusion chromatography of purified 6His-tagged gephyrin and isolated E-domain variants. Full-length gephyrins and isolated E-domains were loaded onto a Superose 6 gel filtration column, and elution of the protein was monitored by measuring absorbance at 280 nm (mAU, milliabsorbance units).

D  Co-immunoprecipitation of HEK293 cell lysates expressing myc-tagged gephyrin together with GFP-tagged gephyrin or G375D using myc-specific antibodies. Control IgG were used to show specificity of the assay.

and thus GABAergic synapse function is severely impaired and (ii) gephyrin-G375D is unable to synthesize the molybdenum cofactor, which is vital for the function of molybdo enzymes. Our findings strengthen previous associations of *GPHN* with epilepsy and other neurodevelopmental disorders (Forstera *et al*, 2010; Lionel *et al*, 2013; Dejanovic *et al*, 2014a) and expand the spectrum to an infantile onset EE. Based on our findings, a severe impairment of inhibitory synapses can be expected, and we suggest that the gephyrin mutation leads to hyperexcitability of neuronal networks and consequently seizures and abnormal neurodevelopment. Moreover,

extensive functional analyses of gephyrin-G375D provide further insights into the molecular mechanisms of gephyrin–receptor interaction and its function at inhibitory synapses.

While oligomerization of gephyrin was found to be intact, we showed a reduced binding affinity of gephyrin-G375D to GABA$_A$Rs and GlyRs. Direct interaction of gephyrin and GABA$_A$Rs has just recently been biochemically shown and is still not fully understood (Maric *et al*, 2011, 2014b; Kowalczyk *et al*, 2013). We showed biochemically and in neurons that efficient gephyrin-GABA$_A$R complex formation requires Gly375 of gephyrin. Thus, besides the

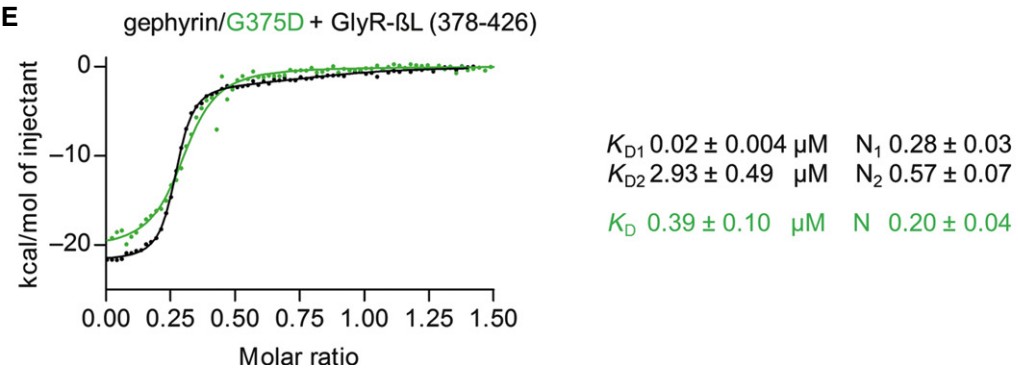

**A** N ... C

363 456

373
KVPEALEMKKKTPAAPAKKTSTTFNI
VGTTYPINLAKDTEFSTISKGAAPSA
424

**B** 6His-gephyrin    6His-G375D

116
66

Input
Pulldown

beads GABA$_A$R α3

$P = 2.4 \times 10^{-06}$

Relative amount of gephyrin

100
50
0

6His-gephyrin
6His-G375D

**C**    6His-gephyrin + α3    6His-G375D + α3

Response units
2 500
2 000
1 500
1 000
500
0

Time (s)    0  200  400  600

**D**

Response units
2 500
2 000
1 500
1 000
500
0

$K_D = 3.2$ μM
$K_D = 10.0$ μM

Gephyrin concentration (μM)    0  10  20  30  40

**E**    gephyrin/G375D + GlyR-ßL (378–426)

kcal/mol of injectant
0
−10
−20

Molar ratio    0.00  0.25  0.50  0.75  1.00  1.25  1.50

$K_{D1}$ 0.02 ± 0.004 μM    $N_1$ 0.28 ± 0.03
$K_{D2}$ 2.93 ± 0.49  μM    $N_2$ 0.57 ± 0.07

$K_D$ 0.39 ± 0.10  μM    $N$ 0.20 ± 0.04

**Figure 6. Gephyrin-G375D has a significantly decreased binding affinity toward GlyRs and GABA$_A$Rs.**

A  Schematic representation of GABA$_A$R α3-subunit with residues of the cytoplasmic loop that were synthesized and used as a biotinylated peptide for downstream applications. Gephyrin core-binding motif is shown in red.

B  Representative pull-down experiment and quantification of 6His-tagged gephyrin and G375D using immobilized α3-peptide. 6His-tagged gephyrin variants were expressed in *E. coli* cells, purified to homogeneity, and incubated with α3-peptides immobilized to NeutrAvidin beads. $n = 3$, normalized to 6His-tagged gephyrin. 38.9 ± 1.5% G375D were pulled down. Analyzed by *t*-test.

C  Representative SPR sensograms of 6His-gephyrin and 6His-G375D binding to immobilized GABA$_A$R α3-peptide. Response units are plotted against time. Injected gephyrin concentrations were 0.5, 2, 10, 25, and 40 μM.

D  SPR binding isotherms of a representative experiment as in (C). Calculated $K_D$ values of 3.2 and 10 μM for gephyrin and G375D, respectively, are indicated by vertical lines. The experiment was repeated three times with two independently purified protein batches and two sensor chips.

E  Representative ITC binding isotherms of the cytoplasmic loop of the GlyR β-subunit (GlyR-βL, residue 378–426) and 6His-tagged gephyrin or G375D. The GlyR-βL-gephyrin binding data (black) were fitted in a two-side model, whereas G375D binding could only be fitted to a single binding site. Dissociation constants $K_D$ and stoichiometry ($N$ = number of binding sites) are given for each binding component. $n = 5$ from two independently purified protein batches. Results are expressed as mean ± SEM.

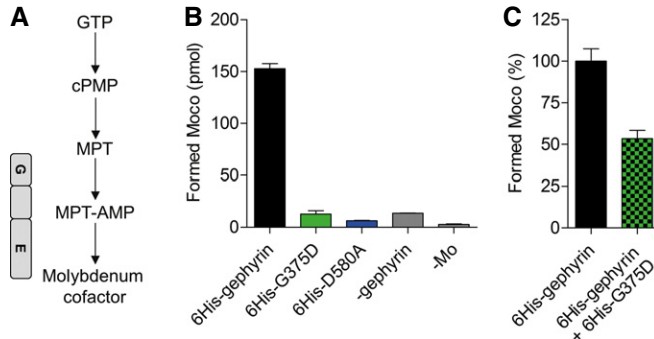

**Figure 7. Gephyrin-G375D is catalytically inactive and unable to synthesize MoCo.**

A   Biosynthesis pathway of MoCo with all stable intermediates. Gephyrin G- and E-domain catalyze the last two steps as depicted.

B   *In vitro* MoCo synthesis assay using 150 pmol purified 6His-tagged gephyrins. D580A is an activity-deficient gephyrin mutation previously identified in MoCo-deficient patients (Reiss *et al*, 2011). Assays without the addition of gephyrin (-gephyrin) or molybdenum (-Mo) served as internal controls of the assay (see also Belaidi & Schwarz, 2013).

C   Experiment as in (B) but with pre-incubation of 6His-gephyrin/G375D in a 1:1 ratio to simulate the heterozygous mutation. *n* = at least 3 with two independently purified protein batches.

Data information: Results in (B, C) are expressed as mean ± SEM.

described common receptor-binding site (Fig 1C), additional regions of gephyrin are necessary for enhanced binding affinity of receptors, as already suggested by others (Maric *et al*, 2011, 2014b). Our thermodynamic studies furthermore showed a bimodal binding of GlyR-βL and WT-gephyrin, containing a high-affinity and a low-affinity binding site (Schrader *et al*, 2004; Specht *et al*, 2011; Herweg & Schwarz, 2012). As the low-affinity binding site is lost in gephyrin-G375D, we conclude that Gly375 is likely an integral part of this binding site.

In neurons, the decreased receptor-binding affinity resulted in diminished gephyrin cluster formation with mutant gephyrin exerting a dominant-negative effect on WT-gephyrin clustering and a significantly decreased size of postsynaptic GABA_ARs clusters. Accordingly, neurons expressing gephyrin-G37D had a decreased GABAergic inhibition, reflected by smaller and less frequent mIPSCs. This suggests an interdependent structural function of the receptors and gephyrin, where not only postsynaptic clustering of GABA_ARs requires efficient binding to gephyrin, but also conversely gephyrin cluster formation is dependent on the efficient binding to GABA_ARs. In agreement with this finding, deletion of the gephyrin-associated GABA_AR subunits α1, α2, α3, or γ2 leads to loss of postsynaptic gephyrin clusters *in vivo* (Schweizer *et al*, 2003; Kralic *et al*, 2006; Panzanelli *et al*, 2011). Congruently, following postsynaptic long-term potentiation of inhibition, which requires the synaptic recruitment of gephyrin and GABA_ARs, receptors and gephyrin are simultaneously accumulating at synapses (Petrini *et al*, 2014). Thus, we propose that the intrinsic property of gephyrin–GABA_AR interaction is essential for their postsynaptic clustering and GABAergic synapse formation. To this end, it is very likely that the G375D substitution ultimately decreases the interaction of all relevant gephyrin–receptor complexes.

In addition to the synaptic dysfunction, we showed that gephyrin-G375D is unable to synthesize MoCo. Structural studies of MoeA, an *E. coli* protein homologous to the gephyrin E-domain, identified the putative active center of the enzyme. It harbors

several highly conserved residues including the glycine corresponding to gephyrin Gly375 (Xiang *et al*, 2001). A mutation of the residue immediately upstream of this glycine abolished the activity of MoeA (Nichols *et al*, 2007). Together with our data, this suggests that Gly375 is part of the active site of gephyrin and essential for the final step in MoCo synthesis (Llamas *et al*, 2006).

Previous studies have shown that heterozygous exon-disrupting deletions in *GPHN* are a risk factor for epilepsy (Lionel *et al*, 2013; Dejanovic *et al*, 2014a). However, they have an incomplete penetrance and their expression strongly depends on the genetic background (Lionel *et al*, 2013; Dejanovic *et al*, 2014a). Similarly, heterozygous gephyrin knockout mice have no obvious acute behavioral phenotype (Feng *et al*, 1998). Some exon-disrupting deletions in IGE patients result in the expression of truncated gephyrin variants that, similarly to gephyrin-G375D, act dominant-negatively on postsynaptic gephyrin clustering (Dejanovic *et al*, 2014a). Compared to these IGE patients, our patient has a much more severe phenotype suggesting that gephyrin-G375D has a more deleterious effect than truncating mutations. A possible explanation is that the properly folded and stably expressed mutant gephyrin probably has the ability to interact with all its regular binding partners. Thus, it has the potential to decrease the clustering and concentration of a number of postsynaptic proteins other than GABA_ARs and GlyRs, thereby acting dominant-negatively on downstream pathways. This would most likely impede plasticity and regulation of gephyrin scaffolds, which are essential for normal development of neuronal inhibitory circuits (van Versendaal *et al*, 2012; Dejanovic *et al*, 2014b; Petrini *et al*, 2014).

The reduced amount of cell-surface expressed GABA_ARs in the presence of gephyrin-G375D exerts a functionally convergent effect similar to that observed by mutations of genes encoding core components of GABAergic postsynapses, for example, GABA_AR subunits and collybistin. Mutations in several GABA_AR subunit-encoding genes, including the synaptic subunits *GABRA1*, *GABRB3*, and *GABRG2* that associate with gephyrin, have been identified in epilepsy syndromes of different degrees of severity (Macdonald *et al*, 2012; Reinthaler *et al*, 2015). Furthermore, mutations in *ARHGEF9* encoding the protein collybistin have previously been linked to seizures and EEs (Harvey *et al*, 2004; Kalscheuer *et al*, 2009). On the molecular level, mutations in these postsynaptic proteins lead to reduced GABAergic signaling, which in turn leads to a lack of neuronal inhibition promoting epileptogenesis. Although dysfunction of gephyrin probably also affects GlyR clustering, this seems to have no significant clinical consequences as the typical symptom of glycinergic defects, hyperekplexia, is missing as already observed in patients with heterozygous *GPHN* deletions (Lionel *et al*, 2013; Dejanovic *et al*, 2014a).

Genetic screenings of *GPHN* in follow-up cohorts of patients with different types of EEs, including Dravet syndrome, did not lead to the identification of additional mutations. Therefore, *GPHN* mutations seem a rare cause of EEs. Nevertheless, the identification of the *de novo GPHN* mutation in a patient with a Dravet-like syndrome is intriguing, since the GABAergic synapse is a well-known major pathway for this phenotype: mutations in *SCN1A*, mainly expressed in inhibitory GABAergic interneurons, *GABRA1,* and *GABRG2*, all have previously been linked to Dravet syndrome (Hirose, 2014). The discovery of a mutation in *GPHN* is also in agreement with the observation that *de novo* mutations in EEs mainly affect genes

involved in synaptic transmission (EuroEPINOMICS-RES Consortium *et al*, 2014).

Lack of MoCo and hence decreased activity of molybdo enzymes in humans, whose function is essential for most forms of life, might additionally contribute to the severe symptoms of the patient. MoCo deficiency indeed shows some overlap with EE, as it is accompanied by severe seizures and rapidly progressive encephalopathy. So far, MoCo deficiency has only been described in patients with a complete loss of MoCo due to recessive mutations in genes encoding proteins involved in the biosynthesis pathway, including two patients with homozygous *GPHN* mutations (Reiss *et al*, 2001, 2011). Parents of these two patients carried a heterozygous mutation and were asymptomatic. Similarly, urinary analysis in IGE patients with a *GPHN* mutation revealed excretion of normal levels of MoCo-dependent metabolites (Lionel *et al*, 2013; Dejanovic *et al*, 2014a). Thus, the presence of one normal *GPHN* allele seems to be sufficient for normal activity of molybdo enzymes. However, we cannot exclude that during critical neurodevelopmental periods, alterations in molybdo enzyme activity might impact neuronal circuits and further lower the threshold for seizures. As loss-of-function MoeA mutants show an increased binding of MoCo (Nichols *et al*, 2007), gephyrin-G375D might decrease the amount of available MoCo in the organism and thereby reduce the activity of molybdo enzymes. We could, however, not dispose of patient's biological samples to proof this hypothesis.

In summary, we present a patient with Dravet-like syndrome and a *de novo* mutation p.G375D in *GPHN* that impairs both the synaptic and metabolic function of gephyrin. Our study strengthens previous associations of *GPHN* with epilepsy and expands the phenotypic spectrum with EEs. Novel compounds, like the recently introduced dimeric peptides that modulate gephyrin–receptor interaction (Maric *et al*, 2014a), might be promising tools to regulate GABAergic postsynaptic stability in patients with gephyrin and GABA$_A$R dysfunction. A decrease in MoCo synthesis could be compensated by administration of the precursor cyclic pyranopterin monophosphate (cPMP), which is the first available therapy for patients with MoCo deficiency (Veldman *et al*, 2010; Schwahn *et al*, 2015).

# Materials and Methods

### Human subjects

Parents or the legal guardian of each patient included in this study signed an informed consent form for participation. This study was approved by the ethical committee of the local institutes. All experiments conformed to the principles set out in the WMA Declaration of Helsinki and the Department of Health and Human Services Belmont Report.

### Whole exome sequencing and data analysis

Whole exome sequencing was performed on blood-extracted DNA of a patient with Dravet-like syndrome and his healthy parents at the Wellcome Trust Sanger Institute (Hinxton/Cambridge, UK). This patient was part of a larger cohort of patients with Dravet(-like) phenotypes that were selected for WES as part of the EuroEPI-NOMICS-RES project. As in approximately 80% of patients with

Dravet syndrome an *SCN1A* mutation is found, *SCN1A* mutations had to be excluded upfront. 3 µg of genomic DNA was sonicated to fragments of 150–200 bp and purified. The exome was then captured using the SureSelect Human All Exon 50 Mb kit (Agilent), and next-generation sequencing (NGS) was performed on a HiSeq2000 (Illumina) platform as 2 × 75 bp paired-end reads. Mapping of the reads to the reference genome (hg19, UCSC Genome Browser) was done using Burrows-Wheeler Aligner (BWA) (Li & Durbin, 2010).

*De novo* variants were called using DeNovoGear (DNG) (Conrad *et al*, 2011), and the generated list of variants was further filtered according to the following criteria: (i) read depth in all individuals ≥ 8; (ii) allele balance in the patient between 0.25 and 0.75 and in the parents ≥ 0.95; (iii) exclusion of variants in tandem repeats and segmental duplications; (iv) posterior probability of *de novo* calling of DNG ≥ 0.5; and (5) exclusion of variants seen in > 1 individual.

We also analyzed the WES data under a recessive model. Hereto, variants were called using Genome Analysis Toolkit (GATK) Unified Genotyper (McKenna *et al*, 2010) and SAMtools (Li *et al*, 2009). For the downstream analysis of the variants, we used GenomeComb (Reumers *et al*, 2012), an in-house developed tool for the annotation and filtering of NGS data. The dataset was further filtered using the following criteria: (i) read depth in all individuals ≥ 8; (ii) allele balance in the patient ≥ 0.95 and in the parents between 0.25 and 0.75 for filtering under a homozygous model and between 0.25 and 0.75 in the patient and his parents for the compound heterozygous model; (iii) exclusion of variants in tandem repeats and segmental duplications; and (iv) a frequency of ≤ 1% in control databases (1000 Genomes Project, Exome Variant Server, in-house data).

### Genetic follow-up screenings

In order to identify additional mutations in *GPHN*, we performed targeted resequencing of *GPHN* using two different gene panels.

(I) For the first gene panel, we designed a MASTR assay (Multiplicom, http://www.multiplicom.com/multiplex-amplification-specific-targets-resequencing-mastr). We used mPCR (Multplicom) to generate primers that amplify the complete coding sequence of the gene as well as flanking intronic regions to be able to identify splice-site mutations. The assay consists of multiplex PCRs followed by a labeling step, both carried out on a Veriti AB machine (Life Technologies). To guarantee an optimal amplification, all amplicons are visually inspected after they have been size-fractionated on an ABI 3730xl DNA Analyzer (PE Applied Biosystems). The different amplicons are subsequently pooled in equimolar amounts, and each sample is labeled with a unique barcode. NGS was then carried out on the pooled amplicons using a MiSeq sequencer (Illumina) as 2 × 300 bp paired-end reads (v3 kit, Illumina). Using this assay, we screened 164 patients with epilepsy and ID. The generated reads were aligned to the reference genome (hg19, UCSC Genome Browser) using BWA, and variants were called using GATK Unified Genotyper (McKenna *et al*, 2010) and SAMTools (Li *et al*, 2009). For the downstream analysis of the variants, we used GenomeComb (Reumers *et al*, 2012).

(II) In a second set of samples, we performed targeted capturing of all exons and 5 bp of flanking intronic sequence of the *GPHN* gene using MIPs as previously described (Carvill *et al*, 2013). Briefly, following hybridization and capturing, sequences are PCR-amplified, incorporating an Illumina adaptor and a sample-specific

barcode to allow sample pooling. Sequencing was performed using a HiSeq platform (Illumina). A total of 124 patients with Dravet (-like) syndrome were screened. After mapping and variant calling, variants were annotated using SeattleSeq. Novel nonsynonymous, frameshift, nonsense, or splicing variants were validated in a second experiment, and segregation analysis was performed when possible. Validation of variants was performed using a second, targeted, independent MIP capture followed by sequencing on a MiSeq (Illumina) or Sanger sequencing or both.

## Data deposition

The EuroEPINOMICS-RES exome sequencing data from this publication has been submitted to the European Genome phenome Archive (https://ega.crg.eu/) and assigned the identifiers EGAS00001000190#, EGAS00001000386#, and EGAS00001000048#.

## Expression constructs

EGFP-tagged and myc-tagged gephyrin and pHluorin-tagged $GABA_A R$ γ2 have been described previously (Dejanovic *et al*, 2014a). 6His-tagged gephyrin and intein-tagged GlyR β-loop *E. coli* expression vectors have been introduced before (Schrader *et al*, 2004; Reinthaler *et al*, 2015). Collybistin II without the auto-inhibitory SH3-domain was cloned into the mCherry-C3 vector (Clontech). Mutations have been generated by fusion-PCR, and all constructs have subsequently been sequenced.

## Cell culture and transfection

Primary hippocampal neurons, HEK293 and COS7 cells were cultured as described previously (Dejanovic *et al*, 2014a). Neurons were usually transfected after 9 days *in vitro* (DIV) using Lipofectamine 2000 according to the manufacturer's manual and cultured for 4 additional days. HEK293 and COS7 cells were transfected with polyethylenimine using standard protocols.

## Immunostaining and quantification of cultured cells

Immunostaining and microscopy of cultured cells have been described previously (Dejanovic & Schwarz, 2014). Briefly, images were taken on a Nikon AZ-C2$^+$ confocal laser scanning microscope as z-stack with three optical sections with 0.5 μm step size. Maximum intensity projections were created and analyzed using NIS Elements 3.2 (NIKON) software. Usually postsynaptic clusters of two 20 × 5 μm regions of interest per neurons were analyzed using the analyze particles option in NIS Elements. Only clusters between 0.09 μm² and 1.5 μm² were considered for analysis. Mean values were compared for significance using Student's *t*-test. Following antibodies were used for immunostaining: anti-gephyrin (1:50, clone 3B11, cell culture supernatant); anti-VGAT (1:500, 131003, Synaptic Systems). Secondary antibodies were goat-raised AlexaFluor 488 and 568 antibodies (1:500, Invitrogen).

## Western blot analysis

For immunoblotting, a standard protocol was followed and detection was performed using chemiluminescence and an electrochemiluminescence system with a cooled CCD camera (Decon Science Tec). The following primary antibodies were used and diluted in Tris-buffered saline/0.5% Tween containing 1% dry milk: anti-gephyrin (1:50, clone 3B11, cell culture supernatant), anti-gephyrin ("Puszta serum" (Smolinsky *et al*, 2008)) anti-GFP (1:1,000, A-11122, Thermo Scientific), anti-myc (1:50, clone 9E10, cell culture supernatant), and anti-mCherry (1:3,000, PA5-34974, Thermo Scientific). Secondary antibodies (1:5,000) were from Santa Cruz Biotechnology.

## Surface immune-labeling of hippocampal neurons

Neurons transfected with myc-tagged gephyrin and pHluorin-tagged γ2 subunit were incubated with a polyclonal GFP antibody (1:200, A-11122, Thermo Scientific, 20% goat serum in PBS) at 4°C for 40 min to prevent internalization of receptors. After extensive washing PBS, neurons were fixed with 4% paraformaldehyde and incubated with Alexa568 secondary antibodies (1:250, Invitrogen, 20% goat serum in PBS).

## Expression and purification of recombinant proteins from *E. coli*

Recombinant 6His-tagged gephyrin and isolated E-domains were expressed in *E. coil* and affinity-purified using nickel-nitrilotriacetic acid resin (Ni-NTA, Thermo Scientific) as suggested by the manufacturer with minor modifications. Cells were lysed in PBS supplemented with 50 mM imidazole and protease inhibitors (Roche) using an EmulsiFlex (Avastin) high-pressure homogenizer. After pelleting cell debris, supernatant was incubated with Ni-NTA beads and affinity-purified using a standard protocol and eluted in PBS containing 250 mM imidazole. After purification, protein was buffer-exchanged to remove imidazole.

GlyR-βL-intein was expressed in *E. coli* ER2566 for 14 h at 18°C and affinity-purified in 300 mM NaCl, 50 mM Tris pH 8.0 containing 1 mM EDTA using the IMPACT protein purification system (New England Biolabs; see Schrader *et al*, 2004). Thiol-induced cleavage of the GlyR-βL-intein fusion protein took place in 150 mM NaCl, 50 mM Tris pH 7.5, 50 mM DTT, and 1 mM EDTA for 24 h at room temperature. The GlyR-βL was separated and enriched using semi-permeable cellulose membrane containing devices (10 kDa and 3 kDa pore sizes) purchased from Millipore and subsequently dialyzed against ITC buffer overnight at 4°C (250 mM NaCl, 10 mM Tris pH 8.0, 1 mM β-mercaptoethanol). All proteins were flash-frozen in liquid nitrogen and stored at −80°C.

## Isothermal titration calorimetry

Isothermal titration calorimetry (ITC) measurements were performed with recombinant 6His-tagged gephyrin or G375D and purified GlyR β-loop residues 378–426 (GlyR-βL). ITC experiments were performed at 25°C using a VP-ITC system with cell concentrations of 20–25 μM gephyrin and syringe concentrations of 200–350 μM purified GlyR-βL. The injection volume was 3–5 μl/3–5 s for each of 50 injections with 240 s spacing and an initial delay of 120 s. The syringe speed was 310 rpm and the reference power adjusted to 5 μcal/s. Raw data were analyzed with Origin 7 software.

## Pull-down assay

A peptide, corresponding to residue 373–424 of the mature GABA$_A$R α3-subunit (KVPEALEMKKKTPAAPAKKTSTTFNIVGTTYPINLAKDTE FSTISKGAAPSA), has been synthesized in house and purified via HPLC (Kowalczyk *et al*, 2013). The peptide was N-terminally biotinylated and carried a hexanoic acid spacer before the first α3-residue. For the pull-down assay, around 2 μM peptide, dissolved in PBS, was incubated with NeutrAvidin beads (Pierce) for 1 h and unbound peptides were washed away with three subsequent washing steps. Peptide-loaded or unloaded beads were incubated with 1 μM purified gephyrin variants for 1 h at room temperature followed by four washing steps. Bound gephyrins were released by boiling in SDS-loading buffer, and proteins were resolved on a 6% SDS–PAGE.

## Surface plasmon resonance spectroscopy

Surface plasmon resonance (SPR) using the biotinylated GABA$_A$R α3-loop (373–424) peptide as ligand was performed with a Biacore-X100 system (GE Healthcare). The peptide was coupled to a streptavidin sensor chip (500 nm thickness; Xantec Bioanalytics) with a target level of 500 response units according to the manufacturer's protocol. Coupling as well as interaction analysis was performed in PBS containing 0.005% Tween-20 and additionally 1 mM β-mercaptoethanol in the measurement. Purified 6His-tagged gephyrin or 6His-tagged G375D were used as analyte in a concentration range of 0–40 μM and a flow rate of 10 μl/min. The relative response units were determined by subtraction of the responses of flow cell 1 from the results of flow cell 2. The sensor chip with coupled peptide was regenerated after each cycle with 1 M NaCl/50 mM NaOH.

## Co-immunoprecipitation

Transfected HEK293 cells were lysed in PBS containing protease inhibitors (Roche) by sonication. Cleared lysate was incubated with myc- or GFP antibody-charged protein A/G sepharose beads (Santa Cruz Biotechnology) for 2 h at room temperature. Control IgG-loaded beads were used to show specificity of the immunoprecipitation. After three washing steps, adsorbed proteins were eluted from the beads by boiling in SDS-loading buffer and analyzed by Western blotting.

## Circular dichroism spectroscopy

The far-UV circular dichroism (CD) spectra from 195–260 nm were recorded at 25°C on a J-715 CD spectropolarimeter (JASCO). Buffer of purified gephyrin was exchanged to 10 mM potassium phosphate, and samples were filtered before measurement. About 0.2 mg/ml of gephyrin were used in a quartz cell of 0.1 cm path length. Spectral acquisition was taken at a scan speed of 10 nm/min, 4 s integration time, and a bandwidth of 1 nm. An average of 10 scans was obtained for all spectra. Data were corrected for buffer contributions and smoothed using the software provided by the manufacturer (System Software version 1.00).

## Gel filtration of gephyrin

Around 1,500 pmol purified gephyrins or isolated E-domains were loaded on a size-exclusion column (Superose 6) connected to an ÄKTApurifier (GE Healthcare) and run at 0.6 ml/min with GeFi buffer (25 mM Tris, pH 7.4, 150 mM NaCl). Elution was monitored by measuring absorbance at 280 nm. Elution times were calculated according to the elution profile of standard proteins of 1,400, 669, 150, and 66.4 kDa.

## MoCo synthesis assay

The *in vitro* MoCo synthesis assay has previously been described in detail (Belaidi & Schwarz, 2013). About 150 pmol *E. coli*-expressed and purified gephyrin was used per condition. Samples without gephyrin or without molybdenum were used as control and show MoCo produced either chemically or by trace amounts of molybdenum in the buffers. Proteins from two independent purifications were used for the assay.

## Electrophysiology

Hippocampal cultures from E19 Wistar rats were prepared and transfected on DIV9 using Effectene (Qiagen) as previously described (Winkelmann *et al*, 2014). Rats were handled and killed according to the permit No. Y9002/15 given by the institutional ethics committee of the Max-Delbrück Center for Molecular Medicine (Berlin, Germany). Transfected neurons were recorded between DIV12 and DIV15 as described (Forstera *et al*, 2010). An EPC-7 amplifier and Patchmaster software (HEKA) were used for patch clamp recordings and data acquisition. Patch pipettes, made from borosilicate glass (Science Products), had resistances of 2–6 MΩ when filled with the intracellular solution containing (in mM) CsCl (130), NaCl (5), CaCl$_2$ (0.5), MgCl$_2$ (1), EGTA (5), and HEPES (30). The standard extracellular solution (pH 7.4) contained (in mM) NaCl (140), KCl (5), MgCl$_2$ (1), CaCl$_2$ (2), HEPES-NaOH (10), and glucose (10). Cells were clamped at a potential of −50 mV. Series resistances ($R_s$), monitored by −5 mV voltage pulses (50 ms) applied every 5 s, were between 5 and 30 MΩ. Experiments with a more than 25% change in $R_s$ throughout the recording were discarded. Data were acquired with a sampling rate of 10 kHz after filtering at 2.8 kHz. Transfected cells were identified according to EGFP fluorescence. GABAergic mIPSCs were recorded in the presence of tetrodotoxin (TTX, 1 μM; Sigma), isolated pharmacologically by blocking NMDA receptors with DL-aminophosphonovaleric acid (APV, 50 μm; Sigma), AMPA receptors with 6,7-dinitroquinoxaline-2,3-dione (DNQX, 10 μM, Sigma), and glycine receptors with strychnine (1 μM; Sigma). Quantitative analysis of mIPSC parameters was performed with an in-house software written in IGOR 6.37A (WaveMetrics) by M. Semtner. Traces of current profiles from 17 different recorded neurons (1 to 5 min long) were analyzed for each condition. The first 50 mIPSCs from an analyzed trace were taken for statistical analysis. Average amplitudes from the experiments were taken for statistics, summarized as mean ± SEM for each condition. The frequency of mIPSCs was determined for each measured cell by dividing the number of events (50) by the analyzed time interval that had passed until 50 events were recorded.

## Statistical analyses

Statistical analyses of biochemical and cellular data were evaluated with GraphPad Prism 5 employing *t*-tests. A *P*-value of < 0.05 was

**The paper explained**

**Problem**

Dravet syndrome (DS) is among the best-described and most extensively studied entities within the epileptic encephalopathies. Up to 80% of DS patients have an *SCN1A* mutation and, although less frequent, mutations in several other genes have also been described. However, a subset of DS patients remains without genetic cause, thereby preventing a genetic etiological diagnosis and hampering opportunities for genetic counseling and therapeutic interventions.

**Results**

We identified a *de novo* mutation in *GPHN* in a patient with Dravet-like syndrome. *GPHN* codes for the moonlighting protein gephyrin that has a structural and modulating function at inhibitory synapses and a metabolic function where it catalyzes the biosynthesis of the molybdenum cofactor. The identified gephyrin-G375D mutation fails to form postsynaptic clusters in neurons and acts dominant-negatively on WT-gephyrin. This affects the architecture of inhibitory postsynapses and signal transmission. We identified a decreased affinity between gephyrin-G375D and the inhibitory neuroreceptors as the potential molecular pathomechanism. Additionally, the mutant gephyrin was unable to synthesize the molybdenum cofactor and activate molybdo enzymes, which might contribute to the phenotype of the patient.

**Impact**

We provide a detailed biochemical, cellular, and functional characterization of a gephyrin mutation in a patient with Dravet-like syndrome. The identification and characterization of this mutation expands our understanding of the molecular mechanisms of epileptogenesis and further highlights that mutations in synaptic proteins cause epilepsy.

considered significant. Significance of average mIPSC amplitudes was tested by one-way ANOVA and *post hoc* Tukey test (IGOR Pro 6.37). Since frequencies of mIPSCs between different experiments were not normally distributed, values are given as median $\pm$ 25$^{th}$/75$^{th}$ percentile, and significance was tested performing the nonparametric Kruskal–Wallis test followed by *post hoc* Mann–Whitney test (IGOR Pro 6.37).

Expanded View for this article is available online.

## Acknowledgements

We thank the patients and their families for their cooperation to this study. We thank Joana Stegemann, Monika Laurien and Veronika Georgieva for technical assistance, Belaidi Abdel Ali and Sita Arjune for help with the MoCo assay, Ines Neundorf for peptide synthesis, and Marcus Semtner and Aline Winkelmann for their experimental support of the neuronal culture electrophysiology. T.D. is a PhD fellow of the Institute of Science and Technology (IWT). A.S. is a postdoctoral fellow of the Research Foundation—Flanders (FWO). This work was supported by the German Science Foundation (DFG SFB635 to G.S.), the program "Investissements d'avenir" ANR-10-IAIHU-06, Wellcome Trust grants 089062 and 098051 (A.P.), European Commission Framework Programme 7 (FP7) projects Synsys-242167 and gEUVADIS-261123 (A.P.), Academy of Finland grants 251704 and 263401 (A.P.), the Sigrid Juselius Foundation (A.P.), the US NIH grant HL113315 (A.P.), the Eurocores program EuroEPINOMICS of the European Science Foundation (ESF), the Fund for Scientific Research Flanders (FWO) (P.D.J.), and the University of Antwerp (P.D.J.), the French program "Investissements d'avenir" (ANR-10-IAIHU-06) (S.W.), the Bundesministerium für Bildung und Forschung BMBF (Era-Net NEURON II CIPRESS to J.C.M.), and the Deutsche Forschungsgemeinschaft DFG Priority Programme SPP 1784 (ME2075/7-1 to J.C.M.).

## Author contributions

BD, AS, PDJ, SW, and GS designed the study. DC, EuroEPINOMICS Dravet working group, IH and SW performed subject ascertainment and phenotyping. BD and VK performed and interpreted biochemical and functional studies. ITC and SPR were performed by NG, electrophysiology by FH and JCM. TD, MZ, PG, DL, CTM, HCM, IH and AP performed and interpreted genetic studies. BD, TD, SW, and GS wrote the manuscript, and all authors revised it.

## Conflict of interest

The authors declare that they have no conflict of interest.

## For more information

1000 Genomes Project, http://www.1000genomes.org

Burrow-Wheeler Aligner (BWA), www.bio-bwa.sourceforge.net

Clustal Omega, http://www.ebi.ac.uk/Tools/msa/clustalo/

dbSNP, http://www.ncbi.nlm.nih.gov/projects/SNP

ExAc Browser, http://exac.broadinstitute.org

Genome Analysis Toolkit (GATK) Unified Genotyper, https://www.broadinstitute.org/gatk

GenomeComb, http://genomecomb.sourceforge.net

Multiplex Amplification of Specific Targets for Resequencing (MASTR), http://www.multiplicom.com/multiplex-amplification-specific-targets-resequencing-mastr

MutationTaster, http://www.mutationtaster.org

NHLBI Exome Sequencing Project (ESP) Exome Variant Server, http://evs.gs.washington.edu

PolyPhen-2, http://genetics.bwh.harvard.edu/pph2

SAMTools, http://samtools.sourceforge.net

SIFT, http://sift.jcvi.org

UCSC Genome Browser, http://genome.ucsc.edu

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
