## [Review Process File · EMBO Molecular Medicine]

Simultaneous impairment of neuronal and metabolic function of mutated gephyrin in a patient with epileptic encephalopathy

Borislav Dejanovic, Tania Djémié, Nora Grünewald, Arvid Suls, Vanessa Kress, Florian Hetsch, Dana Craiu, Matthew Zemel, Padhraig Gormley, Dennis Lal, EuroEPINOMICS Dravet working group, Candace T. Myers, Heather Mefford, Aarno Palotie, Ingo Helbig, Jochen C. Meier, Peter De Jonghe, Sarah Weckhuysen, Guenter Schwarz

Corresponding author: Guenter Schwarz, University of Cologne

Review timeline:

Submission date:	15 April 2015
Editorial Decision:	28 May 2015
Revision received:	01 October 2015
Editorial Decision:	18 October 2015
Revision received:	26 October 2015
Accepted:	29 October 2015

Transaction Report:

Editor: Céline Carret

1st Editorial Decision

28 May 2015

Thank you for the submission of your manuscript to EMBO Molecular Medicine. We have now heard back from the three referees whom we asked to evaluate your manuscript. Although the referees find the study to be of interest, they also raise a number of concerns that need to be addressed in the next version of your article.

As you will see, all three referees are enthusiastic about the work and additional experiments are suggested to increase the conclusiveness of the paper. I would like to particularly draw your attention to two main points that we find very important for our scope:

1) referee 2 questions the *in vitro* Moco production defects resulting from the gephyrin mutation, given the existence of the 21-years old patient. Explanation for this discrepancy should be experimentally addressed by assessing for example the possibility of any residual enzymatic activity. Measures of metabolic changes should also be done and partial rescue shown upon ROS reduction or lactate shunt treatment.

2) referees 2 and 3 suggest performing electrophysiology recordings and although we agree with referee 3 that ideally this should be done in KI mice, we would be happy if recordings were to be performed in wt and mutant transfected neurons.

We would welcome the submission of a revised version for further consideration with the understanding that the referees' concerns must be fully addressed and that acceptance of the manuscript would entail a second round of review.

Please note that it is EMBO Molecular Medicine policy to allow only a single round of revision and that, as acceptance or rejection of the manuscript will depend on another round of review, your responses should be as complete as possible.

I look forward to receiving your revised manuscript.

***** Reviewer's comments *****

Referee #1 (Remarks):

Dejanovic and colleagues investigate a newly identified de novo missense mutation (G375D) in the gephyrin gene in a patient with epileptic encephalopathy resembling Dravet syndrome. The work addresses routes by which the mutation might impact dysfunction and identifies that the recombinant mutant-protein acts dominant-negatively on the clustering of the endogenous gephyrin in cultured hippocampal neurons. This is due to a reduced binding affinity of the mutant to both GABA(A) and glycine receptors, accompanied by a reduction of GABA(A) receptors at the dendritic surface. In addition, they identify a deficiency of the gephyrin G375D mutant to synthesize Moco, suggesting that the two alternative functions of gephyrin, the formation of inhibitory postsynapses and the catalysis of steps in the biosynthesis of Moco are both affected by the mutation and might cooperatively impact on the pathomechanisms underlying the patient's severe phenotype. The work makes a good case for defining this as a critical loss of function and highlights functional significance in both cell lines and neurons in vitro. The paper is clearly written, provides good explanations and raises interesting issues for mechanisms of synaptic scaffolding. Furthermore, it makes good use of biochemistry, further supporting the cell biological data.

Major comments.

1) On the structural level, the mutation is located in the E-domain of gephyrin. A previous study (Saiyed et al., 2007, J. Biol. Chem. 282, 5625) has shown that the E-domain is involved in dimerization, leading to gephyrin hexamers, due to both G-domain trimerization and E-domain dimerization. In the current study, the authors performed size exclusion chromatography with bacterially expressed and purified gephyrin and indicated that both WT- and G375D-gephyrin eluted at the size corresponding to a trimer. Depending on the buffer-conditions and a possible aggregation of full-length gephyrin, separation of the hexameric protein might be difficult by gel filtration chromatography. To assess the effect of the mutation on full-length gephyrin, blue native page might be a more suitable method to support the author's conclusion that the G375D mutation does not affect the biophysical and biochemical properties of the protein. Alternatively, isolated WT and D375D E-domains might be used in size exclusion chromatography experiment to clearly show that the mutation has no effect on the dimerization properties of this domain.

2) The authors nicely show that the G375D mutation affects (reduces) the binding of gephyrin to GABA(A) and glycine receptors. Mutations in human collybistin, a further gephyrin-interacting protein, have been previously shown to be associated with early infantile epileptic encephalopathies and mental retardation. It would be nice to show whether the G375D mutation affects the binding of gephyrin to collybistin as well.

3) The authors show that the G375D mutation abolishes the synaptic as well the metabolic function

of gephyrin. In order to clarify the impact of the metabolic function of gephyrin on the clustering of GABA(A) receptors at inhibitory postsynapses, authors could perform some of the overexpression experiments shown in Figs. 2 and 4 with the D580A mutant, which selectively impairs the metabolic function of gephyrin. Does overexpression of this mutant lead to similar effects in the surface expression of GABA(A) receptors?

Minor comments.

Figure 1A vs. Figure 3A

The authors show and describe that in neurons Geph-G375D is diffusively distributed and fills all cellular compartments. In contrast, in heterologous cells, a clear difference regarding the cytoplasmic distribution of the mutant, as compared to WT gephyrin cannot be observed. Moreover, in heterologous cells, the mutant seems to be more clustered and apposed to the plasma membrane than the WT recombinant gephyrin. Is this consistently observed in all transfected cells?

Figure 4A

Please increase the intensity of both channels, as they can be hardly seen in the printed version.

Legend to Figure 3A

Non-neuronal cells trasfected with...: Please speficy which cell line has been used (HEK293?).

References

The authors should check the reference list carefully and correct it according to the Journal's policies (max. nr. of coauthors, annotations). See: The Journal of neuroscience: the official journal of the Society for Neuroscience...

Referee #2 (Comments on Novelty/Model System):

The authors of the manuscript Dejanovic et al., "Simultaneous impairment of neuronal and metabolic function of mutated gephyrin in a patient with epileptic encephalopathy" use a combination of biochemistry and morphology to characterize the identified gephyrin de novo mutation G375D. The biochemistry and morphology experiments are performed well with relevant statistics.

There have been other publications mainly from the same group identifying gephyrin deletions and spicing defects associated with epilepsy. The mutation identified in the present study is a de novo mutations identified in one patient from a cohort of 143 patient samples. The parents are healthy and the current study fails to provide a clear understanding of what cause Dravet like symptoms in this patient. The evidence showing MoCo defects in vitro should mean that the patient should not have survived past the first few months, however he has managed till 21yrs of age, abliet with epilepsy associated developmental defects.

The evidence shows reduced GABAAR and GlyR binding for gephyrin G375D mutation in vitro; however, a loss of Glycinergic transmission and GABAergic transmission should have caused more severe defects in this patient, which is certainly not the case. Possibility for other compensatory subunit upregulation could expain this. However this angle is not explored and offers little in terms of medical explanation for the phenotype.

Due to these limitations the current study is of low interest to nonspecialists or even neuroscientists in general.

Referee #2 (Remarks):

Dejanovic et al., present a study titled "Simultaneous impairment of neuronal and metabolic function of mutated gephyrin in a patient with epileptic encephalopathy". They use a combination of biochemistry and neuron morphology to determine the mechanistic basis for gephyrin G375D de

novo mutation associated Dravet like symptom in one patient from a cohort of 143 subjects. The study is well conducted with very clear biochemical analysis and morphological characterization, and in the same time also lacks few key experiments that will provide evidence to support their functional interpretation.

1. The data shows clustering defects of gephyrin G375D mutation in mouse hippocampal neurons. The data also shows reduced surface gamma2 GABAAR expression in G375D expressing neurons. The authors connect this to reduced neuronal inhibition. They need to demonstrate that the neurons have altered intracellular Chloride levels and/or are hyperexcitable. It is possible that only a subset of gephyrin dependent GABAARs might be affected as a result of G375D mutation. It would be very interesting to identify this minor sub-population highlighting the significance of GABAAR heterogeneity and role in pathology.

2. The gephyrin G375D is deficient for MoCo in vitro, but the patient seems to have lived till 21yrs of age inspite of this defect. The authors fail to offer an explanation for this. The group of Dr. Schwarz has shown that gephyrin dependent MoCo synthesis in the brain happens in the astrocytes and not neurons. Hence, the lack of MoCo synthesis by G375D mutation suggests that the epilepsy in this patient has astrocytic defects and neuronal defects. Can this be ruled out?

3. In the discussion authors suggest that possible change in intracellular signaling due to lack of gephyrin clustering might answer why de novo mutation in single patient without SCN1A mutation could result in epilepsy. This is an important point and should have been explored further. The data shows normal oligomerization of gephyrin D375D mutation in vitro, reduced binding to GABAAR and GlyR subunits, yet diffuse expression in neurons. It is possible that G375D mutation affects interaction with ArhGEF9 gene product collybistin, leading to clustering defects even if the gephyrin self oligomerization is not affected. Is the gephyrin G375D mutation Palmitoylation defective for membrane anchoring? This should have been addressed with appropriate experiments.

Referee #3 (Remarks):

In this study, a huge collaborative effort of several international laboratories identified a single mutation in the gene of the multifunctional protein gephyrin with an exon-wide sequence analysis of a patient with epileptic encephalopathy. Functional analysis revealed an impairment of molybdenum cofactor synthesis and alteration of gephyrin- and GABAA-receptors cluster formation in cultured hippocampal neurons, suggesting that the identified defect at GABAergic synapses is the mechanism underlying the patient's phenotype.

This study in general is very interesting and allows new insights into functional domain organization of gephyrin and putative molecular bases of specific epileptic diseases.

One major result of the study is the identification of a single mutation in the gephyrin gene by deep sequencing.

The major finding of the functional analysis of this gephyrin mutation (causing an alteration from glycine to aspartate at position 375) is the characterization of reduced affinities to the GABAA-receptors 3 subunit binding region (about 3-fold) and the so called -loop of the glycine receptor beta subunit (about 10 fold) to the recombinant mutated gephyrin disclosed by biochemical experiments using recombinant purified gephyrin and synthetic peptide sequences.

These results are in agreement with the description of an reduced size of GABAA-receptors clusters in cultured hippocampal neurons expressing the mutant gephyrin (in addition to the endogenous gephyrin).

Often functional analysis's of mutated synaptic proteins includes electrophysiological characterizations of altered synaptic properties. Therefore, of course, one might ask for such analysis also in this study. On the other hand, the only conclusive experiment in this respect might need to analyze a knock-in mice line carrying the identified mutation, an approach clearly far

beyond the scope of the current study.

Therefore it seems reasonable to abstain for an electrophysiological analysis and to perform some small improvements of the figures showing the morphological analysis of neurons expressing the mutated gephyrin and the biochemical analysis of purified recombinant gephyrins.

1. Fig. 2 describes the determination of the number of recombinant GFP-gephyrin and Geph-G375D clusters in transfected hippocampal neurons. The authors should indicate how Geph-G375D is detected. Likely as GFP-gephyrin-mutant and therefore should be termed GFP-Geph G375D in the figure.

2. Fig. 2A reveals different classes of gephyrin clusters (large and bright ones, several other smaller ones). Please indicate which threshold settings were used for quantification and whether or not different classes of gephyrin clusters are identical in respect to be apposed to VIAAT puncta.

3. Fig. 2C does not allow to detect the single gephyrin puncta in the enlargement. This should be shown with images of the single channels in addition to the merged image. Please give also the size bar in the enlargement.

4. Fig. 3C is difficult to understand. The authors should show again the single channels and should quantify the degree of overlap of GFP-Geph-G375D and endogenous clusters. In Fig 2 it was shown that most non-mutated (endogenous) gephyrin clusters are localized synaptically (or at least apposed to VIAAT puncta) however GFP-Geph-G375D are mostly extrasynaptically. Assuming that the reduced number of endogenous gephyrin clusters upon GFP-Geph-G375D expression are still mostly localized synaptically, this would imply that they not all can harbour GFP-Geph-G375D. In addition, the quantification of VIAAT apposed endogenous gephyrin clusters should be shown. Please give also the size bar in the enlargement.

5. Fig. 4 reveals a putative functional link of the mutated gephyrin and patient phenotyp showing that GABAA-receptors containing the $\alpha 2$ subunit are not altered in numbers in transfected cultured neurons, however these clusters revealed a reduced size. Assuming that most identified surface GABAA-receptors are localized synaptically, they might interact with gephyrin clusters which do not contain mutated gephyrin (see results from Fig. 3C), thus, the reduced size of GABAA-receptors might be directly correlated with the reduced size of endogenous gephyrin clusters shown in figure 3C. Therefore the authors should discuss in more detail a model explaining the functional implication of the measured reduced affinity and observed alteration in GABAA-receptors cluster sizes. Moreover other earlier studies (Papadopoulos et al., 2007) showed that a reduced number of gephyrin clusters were correlated with a reduced number of GABAA-receptors clusters in hippocampal neurons, however in this study a reduced number of gephyrin clusters (about 50% reduction) is correlated with no reduction in number of GABAA-receptors.

6. Fig. 5A is shown to demonstrate that both GFP-gephyrin variants are similar stable in HEK293 cells. The authors should indicate the antibody used in this WB. For to demonstrate that no degradation products are detectable different other anti-gephyrin antibodies should be used in this experiment. Moreover, the authors should explain why only gephyrin expressed and purified from *E. coli* and not other (eukaryotic) cells are used in these experiments. The authors had published more complete studies earlier (J. Herweg and G. Schwarz, 2012) demonstrating that the expression systems has profound impact on the oligomerization behavior of gephyrin.

7. Fig. 6: Please indicate in this figure the type of recombinant gephyrin in the figure (for example 6His-gephyrin). As in different experiments different recombinant tagged variants (GFP, myc, 6xHis) are used, it should be obvious from the figure and not the legend only, that it is not wild type gephyrin but a tagged-version which is used. Moreover, a short description of all recombinant gephyrin proteins should be given in the method section.

In addition, the authors should discuss the measured affinities in respect to those published from the same group earlier (J. Herweg and G. Schwarz, 2012). Moreover, they should discuss the obvious limitation as gephyrin expressed in SF9-cells does not only revealed different binding affinities (possibly due to posttranslational modifications) to the GlyR beta loop but also different oligomerization behavior (hexamer versus trimer)

We thank you and the reviewers for reading our manuscript. The comments were very helpful to improve the quality of the manuscript. We have addressed all reviewers' concerns in our point-by-point reply below. In the revised manuscript, we have included new results and modified the main text accordingly.

Point-by-point response to the reviewers' questions:

Comments by Referee #1

1) On the structural level, the mutation is located in the E-domain of gephyrin. A previous study (Saiyed et al., 2007, J. Biol. Chem. 282, 5625) has shown that the E-domain is involved in dimerization, leading to gephyrin hexamers, due to both G-domain trimerization and E-domain dimerization. In the current study, the authors performed size exclusion chromatography with bacterially expressed and purified gephyrin and indicated that both WT- and G375D-gephyrin eluted at the size corresponding to a trimer. Depending on the buffer-conditions and a possible aggregation of full-length gephyrin, separation of the hexameric protein might be difficult by gel filtration chromatography. To assess the effect of the mutation on full-length gephyrin, blue native page might be a more suitable method to support the author's conclusion that the G375D mutation does not affect the biophysical and biochemical properties of the protein. Alternatively, isolated WT and D375D E-domains might be used in size exclusion chromatography experiment to clearly show that the mutation has no effect on the dimerization properties of this domain.

ANSWER: We would like to thank the reviewer for the comments. We have followed the reviewer's suggestion and performed size-exclusion chromatography with isolated WT and G375D E-domains. Both elute at a volume corresponding to an E-domain dimer.

2) The authors nicely show that the G375D mutation affects (reduces) the binding of gephyrin to GABA(A) and glycine receptors. Mutations in human collybistin, a further gephyrin-interacting protein, have been previously shown to be associated with early infantile epileptic encephalopathies and mental retardation. It would be nice to show whether the G375D mutation affects the binding of gephyrin to collybistin as well.

ANSWER. This important control is now included in the revision. The mutation does not affect gephyrin-collybistin interaction as shown by co-immunoprecipitation and co-localization studies. Collybistin II is able to induce submembrane Gephyrin-G375D microclusters in COS7 cells. We have noted that the percentage of microcluster-forming cells is slightly decreased for G375D as compared to WT-Gephyrin, which, however, does not reproduce the severe neuronal phenotype.

3) The authors show that the G375D mutation abolishes the synaptic as well the metabolic function of gephyrin. In order to clarify the impact of the metabolic function of gephyrin on the clustering of GABA(A) receptors at inhibitory postsynapses, authors could perform some of the overexpression experiments shown in Figs. 2 and 4 with the D580A mutant, which selectively impairs the metabolic function of gephyrin. Does overexpression of this mutant lead to similar effects in the surface expression of GABA(A) receptors?

ANSWER: Thank you for this interesting suggestion. We generated the D580A variant and found that it was indistinguishable from WT gephyrin in terms of clustering and localization in primary neurons. Thus we conclude that the metabolic and synaptic function of gephyrin are mutually exclusive.

Minor comments.

Figure 1A vs. Figure 3A

The authors show and describe that in neurons Geph-G375D is diffusively distributed and fills all cellular compartments. In contrast, in heterologous cells, a clear difference regarding the cytoplasmic distribution of the mutant, as compared to WT gephyrin cannot be observed. Moreover,

in heterologous cells, the mutant seems to be more clustered and apposed to the plasma membrane than the WT recombinant gephyrin. Is this consistently observed in all transfected cells?

ANSWER: This is not consistently observed in all cells. Gephyrin-G375D as well as WT-Gephyrin form aggregates ('blobs') of diverse sizes in the cytoplasm of non-neuronal cells. However, we have noted that the fraction of cells with submembranous gephyrin microclusters upon Collybistin II expression is slightly smaller compared to WT-Gephyrin (see answer to question 2).

Figure 4A

Please increase the intensity of both channels, as they can be hardly seen in the printed version.

ANSWER: As suggested, we increased the intensity of both channels.

Legend to Figure 3A

Non-neuronal cells transfected with...: Please specify which cell line has been used (HEK293?).

ANSWER: For this image we used COS7 cells and state this now in the figure legend.

References

The authors should check the reference list carefully and correct it according to the Journal's policies (max. nr. of coauthors, annotations). See: The Journal of neuroscience: the official journal of the Society for Neuroscience...

ANSWER: Corrections have been performed.

Comments by Referee #2

Dejanovic et al., present a study titled "Simultaneous impairment of neuronal and metabolic function of mutated gephyrin in a patient with epileptic encephalopathy". They use a combination of biochemistry and neuron morphology to determine the mechanistic basis for gephyrin G375D de novo mutation associated Dravet like symptom in one patient from a cohort of 143 subjects. The study is well conducted with very clear biochemical analysis and morphological characterization, and in the same time also lacks few key experiments that will provide evidence to support their functional interpretation.

1. The data shows clustering defects of gephyrin G375D mutation in mouse hippocampal neurons. The data also shows reduced surface gamma2 GABAAR expression in G375D expressing neurons. The authors connect this to reduced neuronal inhibition. They need to demonstrate that the neurons have altered intracellular Chloride levels and/or are hyperexcitable. It is possible that only a subset of gephyrin dependent GABAARs might be affected as a result of G375D mutation. It would be very interesting to identify this minor sub-population highlighting the significance of GABAAR heterogeneity and role in pathology.

ANSWER: We thank the reviewer for this important point and have followed the suggestions. We performed electrophysiological recordings in neurons expressing Gephyrin-G375D vs WT-gephyrin. As expected from the cellular results, GABAergic amplitude and frequency are reduced in G375-expressing neurons compared to WT-Gephyrin expressing neurons.

Regarding the second point of this reviewer, we agree that it would be interesting to further investigate GABA_AR subtype specific effects. However, this information is even missing for various gephyrin splice variants and dissecting the impact of G375D on specific GABA_AR subtypes would go far beyond the scope of the current work.

2. The gephyrin G375D is deficient for MoCo in vitro, but the patient seems to have lived till 21yrs of age inspite of this defect. The authors fail to offer an explanation for this. The group of Dr. Schwarz has shown that gephyrin dependent MoCo synthesis in the brain happens in the astrocytes

and not neurons. Hence, the lack of MoCo synthesis by G375D mutation suggests that the epilepsy in this patient has astrocytic defects and neuronal defects. Can this be ruled out?

ANSWER: We stated in the results and discussion part that the G375D allele is unable to produce Moco, while the WT allele (still present as the other copy) is probably sufficient to cover the cellular demands of Moco. We also show in the results section (Figure 7), that in contrast to the synaptic function, Gephyrin-G375D has no dominant-negative effect on WT-Gephyrin Moco synthetic activity in vitro.

3. In the discussion authors suggest that possible change in intracellular signaling due to lack of gephyrin clustering might answer why de novo mutation in single patient without SCN1A mutation could result in epilepsy. This is an important point and should have been explored further. The data shows normal oligomerization of gephyrin D375D mutation in vitro, reduced binding to GABAAR and GlyR subunits, yet diffuse expression in neurons. It is possible that G375D mutation affects interaction with ArhGEF9 gene product collybistin, leading to clustering defects even if the gephyrin self oligomerization is not affected. Is the gephyrin G375D mutation Palmitoylation defective for membrane anchoring? This should have been addressed with appropriate experiments.

ANSWER: In the revised manuscript we show new data regarding the analysis of the interaction between the Gephyrin and collybistin II and found that Gephyrin-G375D interacts with collybistin II and forms collybistin II-induced submembranous microclusters in COS7 cells (see also answer to question 2 from Reviewer 1).

Comments by Referee #3

In this study, a huge collaborative effort of several international laboratories identified a single mutation in the gene of the multifunctional protein gephyrin with an exon-wide sequence analysis of a patient with epileptic encephalopathy. Functional analysis revealed an impairment of molybdenum cofactor synthesis and alteration of gephyrin- and GABAA-receptors cluster formation in cultured hippocampal neurons, suggesting that the identified defect at GABAergic synapses is the mechanism underlying the patient's phenotype.

This study in general is very interesting and allows new insights into functional domain organization of gephyrin and putative molecular bases of specific epileptic diseases.

One major result of the study is the identification of a single mutation in the gephyrin gene by deep sequencing.

The major finding of the functional analysis of this gephyrin mutation (causing an alteration from glycine to aspartate at position 375) is the characterization of reduced affinities to the GABAA-receptors α3 subunit binding region (about 3-fold) and the so called β-loop of the glycine receptor beta subunit (about 10 fold) to the recombinant mutated gephyrin disclosed by biochemical experiments using recombinant purified gephyrin and synthetic peptide sequences.

These results are in agreement with the description of a reduced size of GABAA-receptors clusters in cultured hippocampal neurons expressing the mutant gephyrin (in addition to the endogenous gephyrin).

Often functional analysis's of mutated synaptic proteins includes electrophysiological characterizations of altered synaptic properties. Therefore, of course, one might ask for such analysis also in this study. On the other hand, the only conclusive experiment in this respect might need to analyze a knock-in mice line carrying the identified mutation, an approach clearly far beyond the scope of the current study.

Therefore it seem reasonable to abstain for an electrophysiological analysis and to perform some small improvements of the figures showing the morphological analysis of neurons expressing the mutated gephyrin and the biochemical analysis of purified recombinant gephyrins.

ANSWER: We have now included electrophysiological analyses of hippocampal neurons expressing WT-Gephyrin vs Gephyrin-G375D and show that mIPSC amplitudes and frequency are reduced in the presence of the mutant Gephyrin (see also answer to question 1 from Reviewer 2).

1. Fig. 2 describes the determination of the number of recombinant GFP-gephyrin and Geph-G375D clusters in transfected hippocampal neurons. The authors should indicate how Geph-G375D is detected. Likely as GFP-gephyrin-mutant and therefore should be termed GFP-Geph G375D in the figure.

ANSWER: We thank the reviewer for the helpful comments. The reviewer is right, the results in Fig. 2 are derived from GFP-tagged Gephyrins. We have now included the used tags throughout the manuscript and figure legends.

2. Fig. 2A reveals different classes of gephyrin clusters (large and bright ones, several other smaller ones). Please indicate which threshold settings were used for quantification and whether or not different classes of gephyrin clusters are identical in respect to be apposed to VIAAT puncta.

ANSWER: We have now included the threshold settings in the Materials and Methods part. In general, we detect both the small and the larger bright gephyrin clusters as shown in Fig. 2B. We have not observed that specifically a particular class of cluster is synaptically localized and have thus not quantified this parameter.

3. Fig. 2C does not allow to detect the single gephyrin puncta in the enlargement. This should be shown with images of the single channels in addition to the merged image. Please give also the size bare in the enlargement.

ANSWER: We have now split the channels in this and all other Figures to allow a better detection of the individual channels. Scale bars are now included in all Figures.

4. Fig. 3C is difficult to understand. The authors should show again the single channels and should quantify the degree of overlap of GFP-Geph-G375D and endogenous clusters. In Fig 2 it was shown that most non-mutated (endogenous) gephyrin clusters are localized synaptically (or at least apposed to VIAAT puncta) however GFP-Geph-G375D are mostly extrasynaptically. Assuming that the reduced number of endogenous gephyrin clusters upon GFP-Geph-G375D expression are still mostly localized synaptically, this would imply that they not all can harbour GFP-Geph-G375D. In addition, the quantification of VIAAT apposed endogenous gephyrin clusters should be shown. Please give also the size bare in the enlargement.

ANSWER: We now show single channels including a size bar. As Gephyrin-G375D is able to oligomerize with WT-Gephyrin, we assume that most of the clusters contain both gephyrin variants and that it is a matter of molecular stoichiometry, which Gephyrin population dominates within an oligomer/cluster. By our new cellular and electrophysiological recordings, we show that GABAergic signalling is reduced, but not abolished, suggesting that a population of gephyrin and GABA_ARs are synaptically localized.

5. Fig. 4 reveals a putative functional link of the mutated gephyrin and patient phenotyp showing that GABAA-receptors containing the $\alpha 3\beta 3; 2$ subunit are not altered in numbers in transfected cultured neurons, however these clusters revealed a reduced size. Assuming that most identified surface GABAA-receptors are localized synaptically, they might interteract with gephyrin clusters which do not contain mutated gephyrin (see results from Fig. 3C), thus, the reduced size of GABAA-receptors might be directly correlated with the reduced size of endogenous gephyrin clusters shown in figure 3C. Therefore the authors should discuss in more detail a model explaining the functional implication of the measured reduced affinity and observed alteration in GABAA-receptors cluster sizes. Moreover other earlier studies (Papadopoulos et al., 2007) showed that a reduced number of gephyrin clusters were correlated with a reduced number of GABAA-receptors clusters in hippocampal neurons, however in this study a reduced number of gephyrin clusters (about 50% reduction) is correlated with no reduction in number of GABAA-receptors.

ANSWER: We thank the reviewer for raising this interesting point, which we have addressed in more detail by including the electrophysiological analysis. The reduced mIPSC amplitudes mirror the reduced surface-exposed GABA_AR cluster size. Indeed, it is somewhat controversial if reduced gephyrin molecules reduce only the size of GABA_AR clusters or also the number. *In vivo* studies at least do not see a reduced number of clusters (Fischer et al. 2000). However, we see a reduced GABAergic mIPSC frequency, which points to a smaller number of synaptic GABA_AR cluster. It is considerable that in live-cell immunostaining, clusters that are leaving the synapse have been stained and used for the quantification. Further studies are however needed to address the exact molecular dynamics of gephyrin and GABA_ARs at the synapse.

6. Fig. 5A is shown to demonstrate that both GFP-gephyrin variants are similar stable in HEK293 cells. The authors should indicate the antibody used in this WB. For to demonstrate that no degradation products are detectable different other anti-gephyrin antibodies should be used in this experiment. Moreover, the authors should explain why only gephyrin expressed and purified from *E. coli* and not other (eukaryotic) cells are used in these experiments. The authors had published more complete studies earlier (J. Herweg and G. Schwarz, 2012) demonstrating that the expression systems has profound impact on the oligomerization behavior of gephyrin.

ANSWER: We have now rerun the Western blots with a GFP-specific and a polyclonal gephyrin antibody demonstrating the stability of the proteins. We chose to use the *E. coli* system as this is (still) the preferable expression system in the field and allows better to compare results with published results from others. Furthermore, in Fig. 5D we show that Gephyrin-G375D oligomerizes in eukaryotic cells.

7. Fig. 6: Please indicate in this figure the type of recombinant gephyrin in the figure (for example 6His-gephyrin). As in different experiments different recombinant tagged variants (GFP, myc, 6xHis) are used, it should be obvious from the figure and not the legend only, that it is not wild type gephyrin but a tagged-version which is used. Moreover, a short description of all recombinant gephyrin proteins should be given in the method section. In addition, the authors should discuss the measured affinities in respect to those published from the same group earlier (J. Herweg and G. Schwarz, 2012). Moreover, they should discuss the obvious limitation as gephyrin expressed in Sf9-cells does not only revealed different binding affinities (possibly due to posttranslational modifications) to the GlyR beta loop but also different oligomerization behavior (hexamer versus trimer)

ANSWER: We have included the respective tags in all Figures and listed them in the method section. The measured affinities here and reported earlier by our group (Herweg and Schwarz, 2012) are comparable. We chose *E. coli* expressed gephyrin for the interaction studies as there is no significant difference between *E. coli* and Sf9 expressed gephyrin in terms of GlyR β-loop affinity in the original report by Herweg and Schwarz (K_D of *E. coli* vs Sf9 gephyrin (μM): 0.05 ± 0.02 vs 0.03 ± 0.01 for high affinity and 6.25 ± 2.18 vs 2.66 ± 1.28 for the low affinity site). Thus, for the interaction and binding measurements *E. coli* expressed gephyrin represents a useful tool and allows the expression of much larger quantities of gephyrin protein. Furthermore, to our knowledge all other labs performing similar measurements work with *E. coli* expressed gephyrin, which allows a better comparison of data.

2nd Editorial Decision

18 October 2015

Thank you for the submission of your revised manuscript to EMBO Molecular Medicine. We have now received the enclosed reports from the referees that were asked to re-assess it. As you will see the reviewers are now supportive and I am pleased to inform you that we will be able to accept your manuscript pending final editorial amendments.

Please submit your revised manuscript within two weeks.

I look forward to reading a new revised version of your manuscript as soon as possible.

***** Reviewer's comments *****

Referee #2 (Comments on Novelty/Model System):

In the manuscript Dejanovic et al., the Authors use a combination of molecular biology, biochemistry, electrophysiology and morphology to add a novel human mutation on the scaffolding molecule gephyrin that inhibits its MoCo function and synapse function. Interestingly, using the G375A mutation they show that MoCo function can be uncoupled to the synapse scaffolding function. These are novel and interesting findings that will greatly facilitate in understanding the nuances underlying different forms of epilepsy pathology.

Referee #2 (Remarks):

The Authors have done a series of new experiments that address all the reviewer concerns and have raised the quality of the manuscript considerably. I have no further reservations.

Referee #3:

The authors respond to all demands and fulfill all corrections to my full satisfaction. Therefore I suggest the publication in EMBOMM.